# Extreme longitudinal thermal conductivity and non-diffusive heat transport in isotopic hBN

Cléophanie Brochard-Richard[1], Gaia Di Berardino[1], Etienne Herth[1], Chen Wei [1], Federico Panciera[1], Thomas Poirier [2], James H. Edgar [2], Bernard Gil [3], Guillaume Cassabois [3], Maria Luisa Della Rocca[4], Suman Sarkar[5], Nedjma Bendiab[5], Laëtitia Marty[5], Fabrice Oehler [1], Abdelkarim Ouerghi[1] & Julien Chaste [1] ✉

We measure the temperature profile and investigate the thermal conductivity of suspended monoisotopic hexagonal boron nitride (h[10]BN) heterostructures by combining suspended microbridge technique and Raman spectroscopy. The thermal conductivities exceed 1650 W.m$^{-1}$.K$^{-1}$ at room temperature, significantly higher than in previous reports, highlighting the crucial influence of the measurement conditions on the experimental results. By including more data points, we refine our models beyond the accuracy of conventional approaches. Our results show a striking deviation of thermal transport from the classical diffusion regime described by Fourier's law: while the temperature profiles are linear above 300 K, they become clearly nonlinear below this temperature, indicating a strong non-diffusive heat transport regime. This behavior underscores the need for a new theoretical framework to fully account for heat transport in two-dimensional materials. Ultimately, our findings pave the way for innovative heat dissipation technologies and challenge conventional paradigms in nano-heat engineering. This study establishes a practical framework linking Raman-based temperature mapping, the number of measurement points, and thermal simulations to reliably determine the in-plane thermal conductivity of 2D materials.

Efficient thermal management at the nanoscale is crucial to the performance and reliability of electronic and optical devices. This requires both highly effective thermal conductors and insulators, each serving distinct roles in heat regulation[1] and energy conversion[2–4]. Materials with excellent thermal conductivity and poor electrical conductivity are key to efficient heat dissipation, which is mainly conveyed by phonons[1]. Among them, diamond, boron arsenide, and cubic boron nitride are promising candidates for next-generation CMOS heat

spreader[5] or thermal rectifiers[6], as they offer innovative solutions for thermal regulation in modern nanoelectronics.

Materials with high structural anisotropy offer exciting prospects as heat spreaders, as exhibited naturally within two-dimensional (2D) materials or 2D van der Waals (vdW) heterostructures. Their structural anisotropy is reflected in their thermal transport properties between the in-plane thermal conductivity $k_{2D}$ and out-of-plane directions with thermal conductivity $k_\perp$[5]. Graphene and thin graphite films are

[1]Université Paris-Saclay, CNRS, Centre de Nanosciences et de Nanotechnologies, Palaiseau, France. [2]Tim Taylor Department of Chemical Engineering, Kansas State University, Durland Hall, Manhattan, KS, USA. [3]Laboratoire Charles Coulomb (L2C), UMR 5221 CNRS-Université de Montpellier, Montpellier, France. [4]Université Paris Cité, CNRS, Laboratoire Matériaux et Phénomènes Quantiques, Paris, France. [5]Université Grenoble Alpes, CNRS, Grenoble INP, Institut NEEL, Grenoble, France. ✉e-mail: julien.chaste@universite-paris-saclay.fr

remarkably high 2D in-plane thermal conductors up to 5000 W.m$^{-1}$.K$^{-1}$, mostly due to their exceptionally long phonon mean free paths. Meanwhile, k$_\perp$ in some engineered 2D heterostructures can be as low as 0.009 W·m$^{-1}$·K$^{-1}$ [7], which is close to perfect thermal insulation and comparable to xenon gas. Among 2D materials, hexagonal boron nitride (hBN) holds a special position, as it combines efficient electrical insulation (with a bandgap of 6 eV)[8] and high k$_{2D}$ (1000 W.m$^{-1}$.K$^{-1}$)[9].

Nonetheless, in 2D material, different intriguing phonon transport regimes that deviate from the classical Fourier law emerge at low dimensions and within specific temperature windows[10–12]. Notable examples include the prediction and experimental observation of phonon second sound[13,14], Knudsen minima[15,16] or thermal rectification[6]. quantized thermal conductance[17], and coherent phenomena, such as phonon interference and localization. An intriguing non-Fourier heat conduction regime is the hydrodynamic phonon transport[18]. In this regime, the thermal conductivity k$_{2D}$ is no longer an intensive property of the material. Instead, k$_{2D}$ is defined at the local level, and depends on the sample length and width leading to a major challenge for experimental studies[5,13,19].

Despite these advances, direct and quantitative mapping of thermal transport in 2D materials remains highly challenging. Most Raman-based measurements rely on optical heating, which requires precise knowledge of the absorption coefficient and often leads to large uncertainties in the extracted conductivity values. The source of heat is also the same as the temperature lecture, which limits temperature mapping. Furthermore, very few studies have directly measured spatial temperature profiles along suspended samples subjected to a controlled thermal gradient. In this work, we introduce a custom-built thermal characterization platform that combines Joule-heated microheaters, providing a well-defined, tunable heat flux, with spatially resolved Raman thermometry. It thus provides a unique capability to quantitatively probe non-Fourier and hydrodynamic phonon transport in monoisotopic hBN or other 2D materials. To advance our understanding of non-classical phonon transport regimes, the community must be capable of measuring local thermal conductivity with high precision. The comparison between 2-point and 6-point measurements highlights the minimum conditions required for a reliable modeling of heat transport in 2D materials. This framework provides practical guidance for determining when a simplified approach is valid and when a full temperature mapping is necessary. The methodology can thus serve as a reference for future thermal conductivity studies in graphene and related heterostructures.

While most thermal conductivity studies have been on hBN with the natural distribution of boron isotopes (which is 20% $^{10}$B and 80% $^{11}$B), several have examined the thermal conductivity of monoisotopic hBN, material containing either -100% $^{10}$B or $^{11}$B isotope[20,21]. Monoisotopic h$^{10}$BN has a higher thermal conductivity, since the isotopic disorder is eliminated and is an ideal material for the study of hydrodynamic regime of heat transport[10]. We establish a more comprehensive framework for describing longitudinal heat transport in monoisotopic hBN, to bridge the gap between theory and experiment.

## Results

Optical methods for temperature measurement at the nanoscale include radiation[22,23], Raman spectrometry[7,24–27], thermoreflectance[2] and pump probe experiments[13,14]. Electrical approaches, such as thermocouples[28], thermoresistive sensors, and electrical noise thermometry[29] offer alternative strategies for temperature measurement.

In this study, we investigate the thermal conductivity of a suspended 2D vdW heterostructure based on monoisotopic h$^{10}$BN and WSe$_2$ by accurately mapping its temperature profile. This requires a finely controlled heat flow and a precise mobile temperature sensor with sub-micrometer spatial resolution.

Such thermal mapping of a nanoscale device under heating conditions is a major challenge for conventional techniques. To overcome these challenges, several advanced techniques are available, such as EELS[30,31] or SThM[32,33]. In addition, mechanical techniques utilize thermal expansion, mechanical vibration[34,35], mechanical noise thermometry or vibrational bolometry[36]. Regardless of the method, temperature sensitivity or spatial resolution, comprehensive experimental calibrations are always essential to ensure accuracy and reliability.

These challenges are amplified for out-of-equilibrium experiments with high thermal gradients, where thermal transport can be atypical[37]. In the out-of-plane direction of 2D materials, temperature differences can be tens of degrees Celsius[38]. In plane, Joule heating of a suspended 2D material can reach lateral temperature gradients of several thousands of degrees per micrometer[22,23], however, heat transport is generally governed by hot electron thermal emission, excluding phonon transport[22,23].

To achieve high spatial resolution of lateral phonon transport, a few studies have combined both local thermometry maps with fixed and independent heating sources using microheaters as described in ref. 39. In our study, the heated region is an extended zone, a silicon cantilever, controlled by electric Joule heating. Temperatures were mapped optically by Raman spectroscopy over the entire device area[40]. This method allows thermometry and heat flux measurements, in two-point, four-point and even N-point configurations. From these measurements, we extrapolate high values for the thermal conductivity of hBN. Moreover, heat transport deviates remarkably from the conventional diffusive regime postulated by the Fourier's law, since the expected linear temperature gradient evolves to more complex profiles depending on the experimental parameters.

The experimental set-up, shown in Fig. 1a, consists of a thin 2D van der Waals heterostructure (2D vdW-HS) made of WSe$_2$/hBN, suspended between two silicon (Si) cantilevers. The image is done with open source Blender software. The Si cantilever fabrication has been described in a previous work[41,42]. The two small constrictions on each cantilever act as electrical and thermal resistors. Each microheater is fixed to the silicon substrate and thermalized by the thermal bath at temperature $T_{bath}$. When an electric current pass through the left microheater, the resulting Joule heating increases the temperature of the thick section of the microheater uniformly between the two constrictions to the temperature $T_{hot}$. The WSe$_2$/hBN is suspended across a 9 μm gap between the two microheaters, as shown in Fig. 1b for the sample HS1. Herein, the hBN flake is 126 nm thick and the cantilevers are 3.5 μm thick. We developed a specific transfer method, with a suspended film of PPC, for this delicate step which is described in Supplementary Information.

In the conventional Fourier heat transport regime, the heat flux Q through the 2D vdW-HS between the hot and cold microheaters is determined by the thermal conductivity of the material k and the temperature difference, $\Delta T_{2D} = T_{hot} - T_{cold}$, according to Fourier's law $\mathbf{Q} = -k\nabla\mathbf{T}$

In a first approximation, we can describe the 2D vdW-HS as a membrane of length $L$, width $W$ and thickness t with uniform thermal conductivity $k_{2D}$. The corresponding thermal impedance is simply defined as $Z_{2D} = L/(W \cdot t \cdot k_{2D})$. Similarly, the cold cantilever has a thermal impedance $Z_{ref}$ which is related to the conductivity of silicon $k_{Si}$. If $k_{Si}$ and $Z_{ref}(k_{Si})$ are known, $k_{2D}$ can be extracted from temperature measurements. The same heat flow Q is passing through the two thermal impedances connected in series (Fig. 1c). As with a standard electrical voltage divider, the measurement of $\Delta T_{2D}$ and $\Delta T_{ref} = T_{cold} - T_{bath}$ provides a value of $Z_{2D}(k_{2D})$. In our case, finite element modeling was required to establish properly $Z_{2D}(k_{2D})$ and $Z_{ref}(k_{Si})$, because of the non-ideal shapes in our sample.

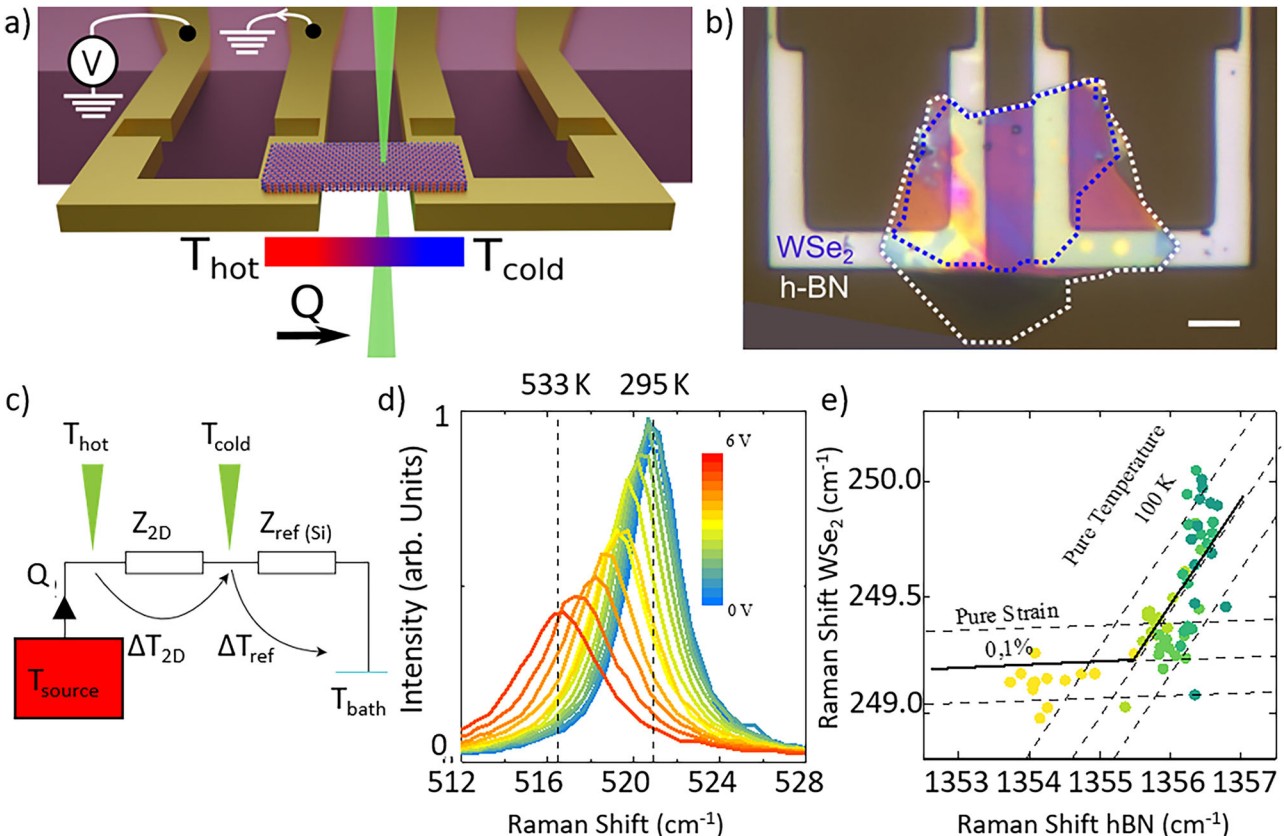

**Fig. 1 | Measurements of thermal conductivity. a** Illustration of the experimental set-up. Two silicon microheaters support a van der Waals heterostructure. One of the microheaters is heated by the Joule effect. The heat flux and temperature map are extracted by confocal Raman spectroscopy at 532 nm. **b** Optical image of sample HS1. Dotted lines serve as a guide for the eye to identify the location of the WSe2 and hBN material (scale bar: 10 μm). **c** Equivalent simplified thermal circuit, showing temperature probes $T_{cold}$ and $T_{hot}$, heat source, thermal bath and two thermal impedances ($Z_{2D}$ and $Z_{ref}$). **d** Raman spectrum of the Si microheater under different bias voltages ($V = 0$ V to 6 V) at room temperature. **e** Diagram of the position of the Raman peak $E_{2g}$ of WSe2 as a function of the position for the Raman peak $E_{2g}$ of hBN along a sample under Joule heating. The two black vectors define two new axes, governed either by pure deformation or by pure heating.

The temperature along the device was measured by Raman spectroscopy. We have meticulously calibrated the temperature for each material, see Supplementary Information.

Heat is generated by one of the microheater (the left one in Fig. 1b). Figure 1d shows the Raman spectral response of the LO peak around 520 cm$^{-1}$ for the left silicon microheater, as a function of the applied electrical input voltage V on the microheater, while the set-up is at ambient temperature. At the maximum power of 26 mW ($V = 6$ V), the first-order Raman Si peak was strongly shifted and from this we estimated a corresponding temperature increase of 238 K. The thermal bath temperature, $T_{bath}$, is the temperature of the left microheater when $V = 0$ V. During the heating experiments, this temperature can be measured experimentally by Raman thermometry by probing the Si substrate close to the cantilevers. This temperature remains constant at $T_{bath}$ when the microheater is subjected to Joule heating in our experimental range.

We also check whether heat losses by thermal radiation and convection, from the hot to the cold cantilever, due to the low pressure in the measuring cell (10$^{-4}$ mbar), stay marginal, presented in Supplementary Information.

In every material, the Raman peak frequency ω is not only temperature-dependent, but also depends on the strain or doping. For 2D materials, such as graphene or MoS2, each contribution can be quantified by constructing a diagram that carefully follows a selected Raman peaks positions[43,44]. Here, we extend this concept to 2D vdW HS of hBN and WSe2.

In the HS1 sample, the monoisotopic hBN flake is 100 times thicker than the WSe2 and expected thermal conductivity of the hBN is 20 times higher. In supplementary Information, modulated thermo-reflectance measurements were conducted on supported natural and isotopic hBN. The thermal conductivity is almost double for the isotopic case, in accordance with a previous report on graphene[45]. The thermal transport is dominated by hBN and the WSe2 is used as a thermal probe. To corroborate this assumption, we have proceeded to multiple analyses presented in Supplementary Information. In Fig. 1e. we analyze the experimental Raman peak positions of the WSe2 $A_{1g}$ peak around 249 cm$^{-1}$ and the $E_{2g}$ peak of h$^{11}$BN around 1356 cm$^{-1}$, along a suspended heterostructure, referred as HS3. It is measured under an out-of-equilibrium temperature with a controlled thermal gradient across the device. The data points follow two different trends representing either a pure mechanical deformation or a pure thermal heating (black lines). The deformation coefficients are extracted from the literature[46–48] and the temperature coefficients are determined experimentally. From this, the strain and the temperature were extracted. The strain variation along our device is quantitively below 0.1% and the Raman peak shift of WSe2 due to strain is negligible. The alignment of the WSe2 and hBN Raman shifts along the calibration curve confirms that WSe2 Raman modes reliably track the hBN temperature even under high-gradient, nonequilibrium regimes. The WSe2 Raman spectrum was easier to measure and is therefore employed as an ideal thermometer for hBN. The accuracy of this hypothesis is supported in the Supplementary Information by measuring multiple

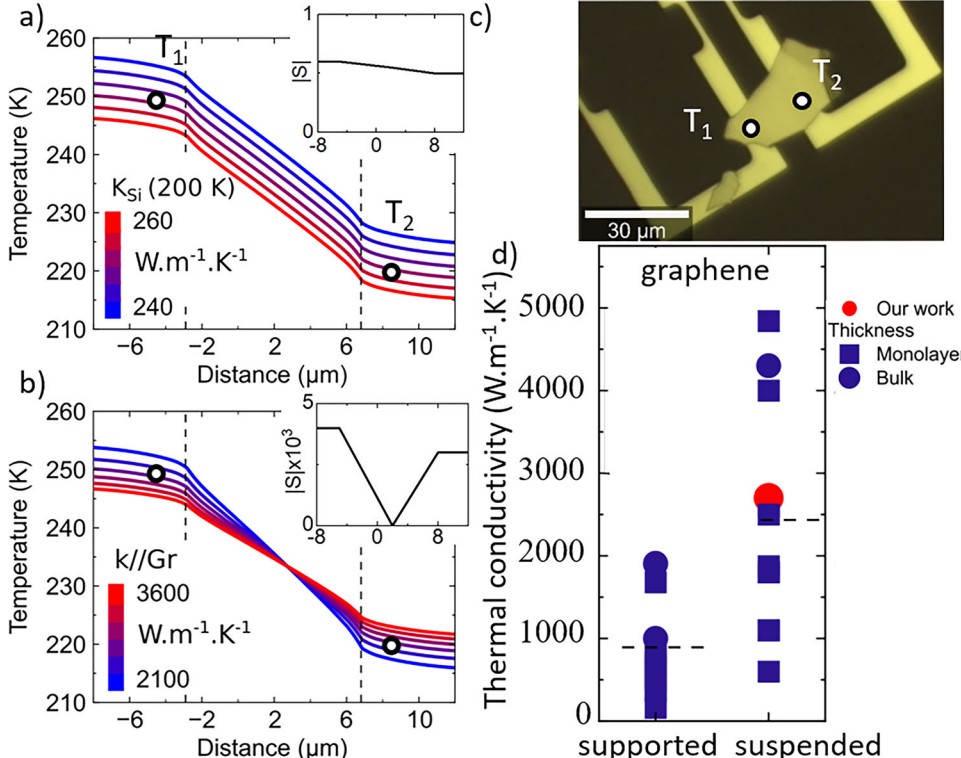

**Fig. 2 | Thermal conductivity of suspended graphite, sample Gr1. a,b** Calculated temperature profiles along a graphite flake for different thermal conductivities of Si and graphite, respectively. Lines are simulation, dots are measured data (**a**) with $k_{2D} = 2700$ W.m$^{-1}$.K$^{-1}$ and (**b**) by fixing $k_{Si}(200K) = 252$ W.m$^{-1}$.K$^{-1}$ and varying $k_{2D}$. The dashed lines delimit the supported (edge areas) and the suspended part (middle area). The dots represent the measured temperature of our sample. In inset the sensitivity $S$ as $S = \partial T / \partial k$. **c** Optical image of the sample **d** Comparison of thermal conductivity measured in different configurations for graphene extracted for the literature. The red points are relative to our work. The horizontal dashed line is the mean value for each column.

points in diverse experimental contexts. In the region in contact with the microheater, hBN is sandwiched between Si and WSe$_2$ and the three materials were thermally anchored.

**Two-point thermal conductivity measurement**

To determine the thermal conductivity requires measuring at least two temperatures: $T_{hot}$ and $T_{cold}$ as in ref. 7. In our case, these readings must be complemented by the measurement of the reference temperature, $T_{bath}$. This method was used to experimentally determine the thermal conductivity of the thin graphite membrane shown in Fig. 2a, b. Here, the graphite membrane is used as a test-bed 2D material where probing the proposed method. We apply a Joule heating power $P = 5.6$ mW and we set the bath temperature to $T_{bath} = 146.15$ K. The temperature differential $\Delta T_{2D}$ between the two measurement points was about 30 K, ($T_{hot} = 251.15$ K, $T_{cold} = 221.15$ K. The temperature difference $\Delta T_{ref}$ between the cold area and the bath is much larger, indicating that the graphite exhibited a higher thermal conductance than the silicon arm. The heat flow is well described by the thermal equivalent circuit shown in Fig. 1c, thus we can determine the values of k$_{2D}$ and k$_{Si}$ from our measurements. To determine the numerical value, we perform finite element simulations (Comsol) of the heat flow based on the actual geometry of the sample. The only free parameters were the silicon thermal conductivity $k_{Si}$ and the graphite thermal conductivity $k_{2D}$.

Figure 2a shows the effect of $k_{Si}$ on the 2D membrane temperature at fixed $k_{2D}$. There is a uniform temperature shift of the temperature profile ($\Delta T_{2D}$ near constant, variable $\Delta T_{ref}$). Alternatively, an increased $k_{2D}$ reduces the temperature gradient across the membrane (i.e. smaller $\Delta T_{2D}$). With this methodology, we obtained the best fit value for $k_{2D} = 2800 \pm 700$ W.m$^{-1}$.K$^{-1}$. In Fig. 2c, we report a non-exhaustive list of the thermal conductivities measured on graphene

and graphite. Our value agrees with the average literature value for the samples based on the suspended membranes and confirming the experimental approach. The temperature noise was determined for the different materials from a statistical analysis of the Raman spectra at different regions of the samples under constant conditions and temperature (see Supplementary Information). The temperature standard deviation is $\delta T_{Si} = 3$ K for silicon, $\delta T_{WSe2} = 5$ K $- 8$ K for WSe$_2$, depending on the method, and $\delta T_{Gr} = 3.5$ K for graphene. The signal is significantly above the experimental noise. Note also that the silicon and graphite temperature are similar, indicating a good thermal anchoring.

To further refine the experimental approach and ensure the reliability of our thermal conductivity measurements, we adopted a multi-point measurement strategy. This method provides a more comprehensive assessment of the temperature distribution and reduces the uncertainties related to local variations, contact resistance and anisotropic heat transport.

**Six-point thermal conductivity measurement**

When the thermal contact impedances are not negligible (for example, in the ballistic regime), a four-point configuration measurement of the thermal conductivity is required. Even in this framework, additional parameters can lead to discrepancies between the model and the experiment. These include the temperature dependence of $k_{2D}$, the anisotropy and the out-of-plane component of $k_{2D}$, the large thermal contact area and the complex geometry of the sample. Note that the required quantity of temperature measurements increases with the complexity of the system.

As verified through full temperature mapping, Fig. 3a−c, only six measurement points are sufficient to accurately reproduce, with COMSOL, the experimental temperature profile across the entire

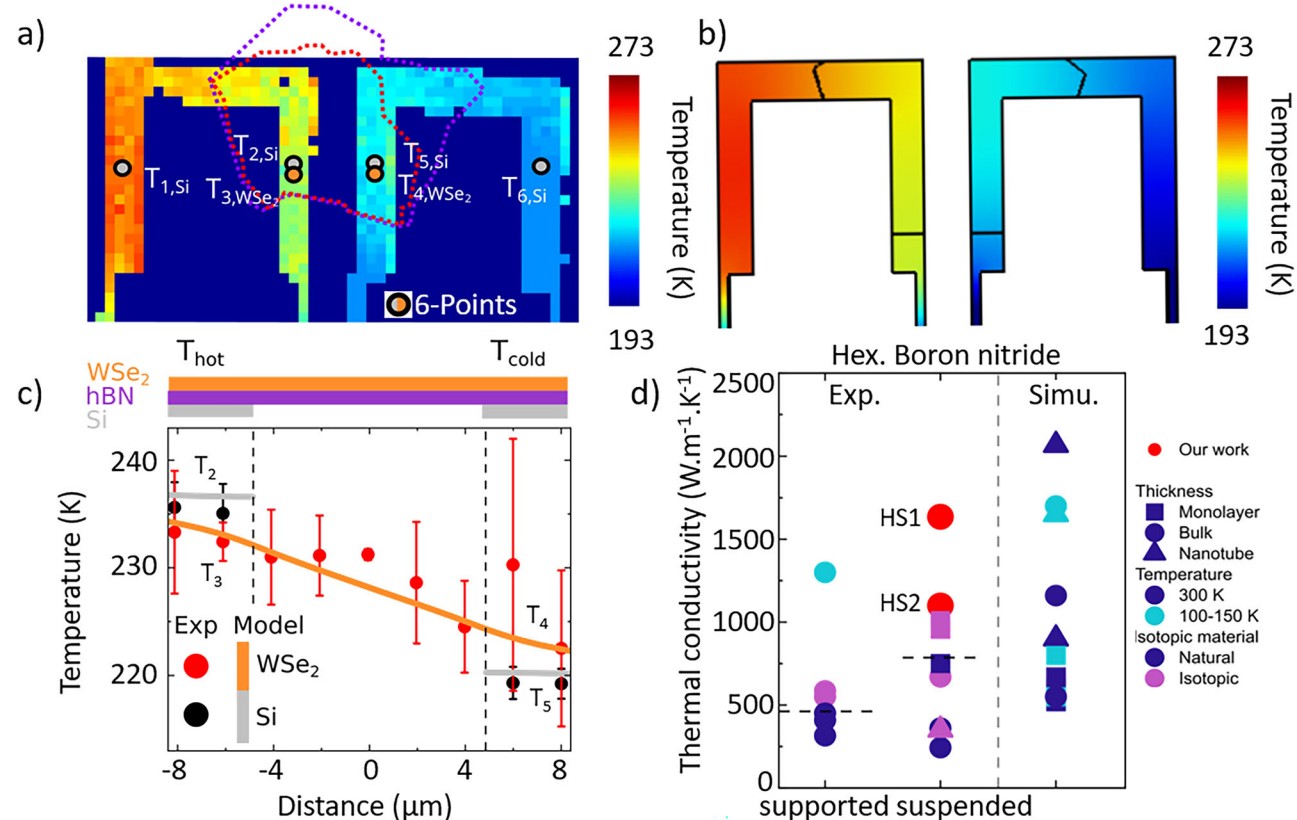

**Fig. 3 | Temperature cartography of the vdW-HS shown in Fig. 1b (HS1).**
**a**, **b** Experimental and simulated temperature mapping of the two cantilevers, respectively. for $T_{bath} = 109\,K$ and $P = 4.6\,mW$. **c** Temperature profile, experimental (point) and simulated (line) for WSe$_2$ and silicon along the vdW-HS made of hBN

and WSe$_2$. **d** Comparison of thermal conductivity measured in different configurations for hBN, extracted from the literature. The horizontal dash line is the mean value for each column.

---

sample when the system operates within the classical heat transport regime. Increasing the number of sampling points does not significantly improve the fit quality or the extracted thermal conductivity This finding provides a practical guideline for optimizing Raman-based thermal measurements, emphasizing the balance between experimental efficiency and physical accuracy. All details about the extraction of thermal conductivity, from these temperatures are presented in Supplementary Information, Section S8–S11. Basically, we minimized the parameter $\Sigma\Delta T^2 = \sum_{points}(T_{mes} - T_{sim})^2$ with simulated $T_{sim}$ and measured temperature $T_{mesured}$ and a strict conditions for each point with ($|T_{mes} - T_{sim}| < \delta T$).

Figure 3a–c illustrates the temperature cartography along the sample presented in Fig. 1b, with $T_{bath} = 109\,K$ and $P = 4.6\,mW$. In Fig. 3c, the red dots and orange line correspond to the temperature data and the simulations, for WSe$_2$, respectively. The error bar was calculated using a mean value derived from five data points in the cross-section. The uncertainty on the WSe$_2$ temperature is higher in the supported region due to a lower Raman intensity (Supplementary Information S14). The black dot and the gray lines represent the data for Si. The temperature of the cantilever in the hot (cold) state is 235 K (219 K). The difference is 16 K, which is slightly higher than the difference $\Delta T_{2D} = 12\,K$. There is a temperature gradient in the 2D heterostructure. In this case, the thermal contact resistance is low but nonnegligible, thereby justifying a four-point measurement. In the COMSOL simulation, the interface is represented by a thin layer (100 nm) at the interface between two materials, with conductivity $k_{interface}$. Given the considerable dimensions of the interface and the complicated nature of the sample, it remains a challenge to find a final solution in

phase space ($k_{2D}$, $k_{Si}$, $k_{interface}$). Two additional measurement points are thus required, located at a greater distance from the 2D material, as indicated by the black circles in Fig. 3c. By using a total of six measurement points (black circle in Fig. 3a), we determined the in-plane thermal conductivity of the heterostructure for sample HS1 was $k_{2D} = 1650 \pm (+550, -350)\,W.m^{-1}.K^{-1}$. For all samples, $k_{Si}(300K) = 124 - 156\,W.m^{-1}.K^{-1}$ agreeing with the literature value (140 W.m$^{-1}$.K$^{-1}$)[49,50]. Figure 3a, b illustrate the temperature mapping of the two cantilever structures, derived from the data and the simulation, respectively. The temperature gradient along the structure is clearly visible and non-trivial. There is a perfect match between the experiment and the model, which justifies a Comsol simulation and a minimum of six measurement points, only the coldest part of the sample is difficult to reproduced within few degrees. We demonstrate in the Supplementary Information, that the measured temperature profile is not affected by residual strain (S5), laser probe heating (S4) and we proceed to analyses and discussion, with uncertainty analyses, along the Supplementary Information for microheater and 2D thermal conductivities (S6, S8), temperature dependence of the thermal conductivity (S9) the hBN out-of-plane thermal conductivity (S10), or interfacial thermal resistance (S11).

As shown in Fig. 3d, our value for $k_{2D}$ of about $1650 \pm 550/350\,W.m^{-1}.K^{-1}$ represents a high value of thermal conductivity compare to the literature with $k_{2D} = [630 \pm 90/65, 585 \pm 80, 751 \pm 340, 1009 \pm 313]\,W.m^{-1}.K^{-1}$ [9,20,21,51]. Recent studies have shown that the thermal conductivity of van der Waals materials strongly depends on whether the samples are supported or suspended, their isotopic purity, thickness, and the

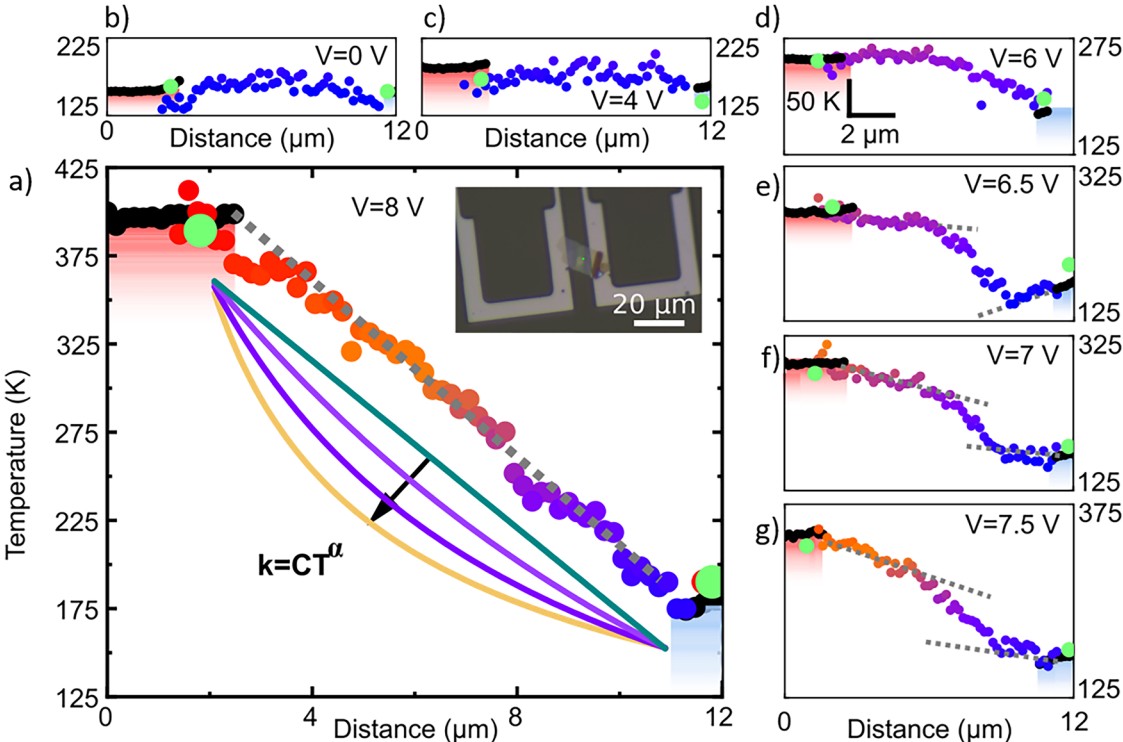

**Fig. 4 | Temperature profile from linear to non-linear for sample HS2.**
**a–g** Temperature of silicon (black) and WSe$_2$ (colored point) along the 2D sample for $V$ = 8, 7.5, 7, 6.5, 6, 4 and 0 V. The green dots are the average temperature of WSe$_2$ in the supported region. Gray dashed lines serve as a guide for the eyes. In (**a**), the four colored lines represent the temperature profile expected in the classical diffusion model for different coefficients $\alpha$ = {0, -1, −2, −3} as $k_{2D}(T) = CT^\alpha$. where the inset shows the optical image of the sample LMM2.

measurement temperature, see Supplementary Information section 12. Suspended samples generally exhibit conductivities two to three times higher than supported ones (Fig. 2c), while isotopically enriched hBN and graphene show up to a twofold increase compared to natural materials[45]. Measurement over larger flakes improve the thermal conductivity within the context of the hydrodynamic regime in 2D material[12]. Thicker flakes also tend to display higher conductivities[52], contrary to earlier assumptions for monolayers, it is still a debate. In our measurements, the exceptionally high value of $k_{2D}$, in HS1, even compared to HS2, is consistent with these trends, reflecting a favorable suspended configuration and high material quality. Since the sample is hBN, an insulator, covered with a monolayer of WSe$_2$, the electric current is only heating the first microheater and we assume that the extracted thermal transport is purely due to phonons.

In the simulation, we optimized the value of $k_{interface}$ equal to 1.4 W.m$^{-1}$.K$^{-1}$ for a corresponding interface thickness of 100 nm. The thickness is optimized to speed up the calculation and does not affect the results. The out-of-plane conductivity of hBN was 5 W.m$^{-1}$.K$^{-1}$, with a value extracted from literature (see Supplementary Information). The found value is confirmed by measurements on the following sample, named HS2.

**From a classical transport with linear temperature profile…**
In this section, we describe the heat transport and thermal conductivity by using an increasing amount of temperature data, and mapping the temperature profile along the 2D material. This approach raises a key question: how much additional insight into the thermal properties can be gained beyond the conventional two- or six-point configuration? To improve the signal-to-noise ratio, we maximize $\Delta T_{2D}$ by measuring a thinner sample with higher thermal impedance. The temperature profile along the vdW-HS for the HS2 sample (22 nm thick), was obtained at $V$ = 8 V as seen in Fig. 4a. Several important conclusions can be drawn directly from the data. The measured $\Delta T_{2D}$

reaches 216 K over a sample length of 8 μm. To the best of our knowledge such temperature gradients has been scarcely reported in the literature[22,23], and is unique in a 2D electrical insulator. Figure 4a shows the raw data and represents a straightforward conversion of the Raman peak position to temperature for each material, without consideration of mechanical deformation. This result validates our measurement methodology, as the temperatures measured for WSe$_2$ and Si are well aligned at the contact region. Furthermore, it provides additional confirmation of low contact thermal resistances and validates the absence of deformation in the structure. A homogeneous tensile extension or compressive strain applied to a 2D material induces a shift in the Raman signature relative to silicon, which is absent in Fig. 4a. The thermal gradient across the suspended heterostructures is quasi-linear, akin to the description of the Fourier regime, and exhibits a temperature dependence of $k_{2D}$ as $k_{2D}(T) = CT^\alpha$. where $C$ and $\alpha$ are fitting coefficients. The exponent $\alpha$ is close to 0, between 0 and -1 which allows a classical treatment of the data in the following analysis analogous to HS1. Both dependence $\alpha$ = -1 and $\alpha$ = 0 have been compared by simulation and it does not affect the extracted thermal conductivity (See Supplementary Information S9). Again, modeling with Comsol, we fitted our data with $k_{2D}$ = 1180 ± 150 W.m$^{-1}$.K$^{-1}$ (corresponding to $\alpha$ = 0). As the temperature difference is large, it is necessary to introduce a temperature dependence of k$_{2D}$ in the Comsol data analysis. Moreover, our data best fit with an out-of-plane thermal conductivity of 5 W.m$^{-1}$.K$^{-1}$ in agreement with literature (see Supplementary Information).

**… to a non-classical regime**
Surprisingly, another phenomenon occurs, visible in Fig. 4b–g when we reduce the applied bias and thus the temperature of the hotspot. The temperature gradually changes from linear to non-linear. The effect is maximum at $V$ = 6.5 V and 7 V where $\Delta T_{2D}$ equals to 140 K and 120 K, respectively. In some flake areas, the temperature remains

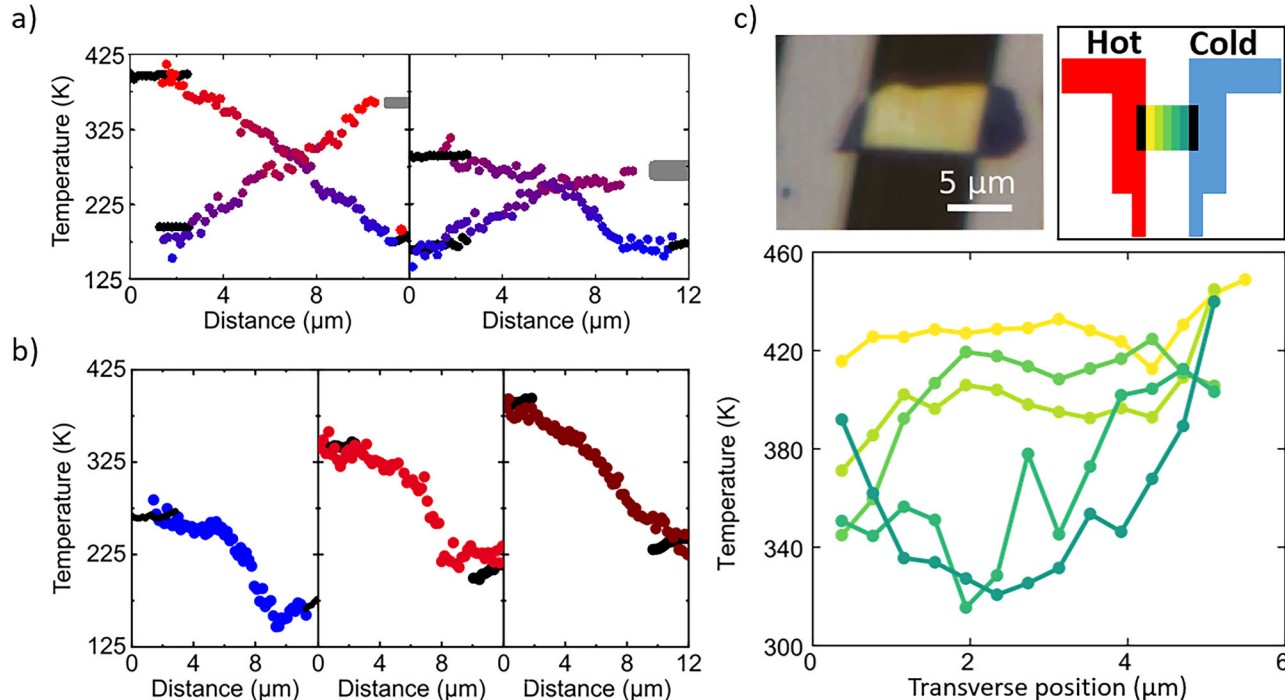

**Fig. 5 | Nonlinear transport. a** Longitudinal temperature profile, along the x-axis, when the direction of heat flow is reversed with respect to the measurement in Fig. 4a, when thermal transport in HS2 shows a linear (left) and nonlinear (right) behavior. **b** Longitudinal temperature profile along the x-axis for different

$T_{bath}$ = 157.15 K, 165.15 K and 186.15 K but the same $\Delta T_{2D}$ for HS2. **c** Optical image of HS3, illustration of the measurement and temperature along the transverse direction for HS3 with the optical image of the sample.

almost constant and then decreases sharply over a few micrometers, with a slope being steeper than in the $V = 8$ V case. At $V = 6.5$ V, the measured temperature along the suspended 2D is even colder than that of the contact. In this case, the phonon transport can no longer be described by the classical Fourier law. As the heat flow is expected to be the same at each position and the temperature gradient is no longer constant, this indicates that the thermal conductivity cannot be defined as an intensive property of the material. In this picture, we can only define the Fourier law locally $k_{2D}(T, x)$, where $x$ is the longitudinal axis. This unusual regime disappears when $V < 4$ V. At $V = 0$ V, Fig. 4b, the temperature is constant, and used as a reference. This confirms that the temperature variation was due to heating. It was possible to see a small non-linearity in the Fig. 3a for HS1 but it is close to the noise. In Fig. 4c–g, for clarity, small global offsets were applied to the 2D temperature in order to align it with the silicon temperature (5 K in average, 15 K at maximum). Anyway, this correction does not affect our interpretation.

In Fig. 5a, the configurations of the microheater and the reference have been reversed in data acquisition on sample HS2, resulting in an opposite direction of the heat flow with respect to the configuration used in Fig. 4. In both the linear and non-linear regimes, the temperature profile was symmetric with respect to the center of the suspended 2D material. This measurement supports the idea that the nonlinearity is due to unusual thermal transport in the 2D flake and not to a localized defect. In Fig. 5a, the temperature of the silicon microheater at 11 μm (gray), for the reversed heat-flow case, was estimated with higher uncertainty and position due to missing data. Figure 5b shows three measurements at different $T_{bath}$ values, with the same $\Delta T_{2D} = 125$ K. As $T_{bath}$ increases, the non-linearity decreases, indicating that it is not uniquely related to the variation of $\Delta T_{2D}$. So far, the temperature profile has been measured along the longitudinal direction (x-axis). In addition, the experimental setup enables the temperature measurement along the transverse direction (y-axis). Figure 5c shows the temperature measurement along the y-axis in a

third sample, referred to as HS3. For this sample, it was possible to resolve the Raman peak of the hBN flake with reasonable accuracy within a reasonable integration time and to optically separate strain and temperature, as shown in Fig. 1e. There is a pronounced temperature variation in the region of the sample close to the cold electrode along the transverse direction, which has a nearly parabolic profile. In the center of the suspended flake, the temperature is about 318 K, while at the edges, it rises to 393 K - 423 K. This measurement further confirms a deviation from the classical heat transport regime in the suspended hBN samples.

## Discussion

Considering previous works and theoretical expectations for thermal transport in isotopic hBN, together with the range of measured temperatures in our experiment (between 125 K and 325 K in Fig. 4e, f), we expect the sample to exhibit phonon transport regime in the Ziman one. In this regime, the coefficient α is close to -1 ($k_{2D}(T) = CT^\alpha$ and normal scattering events dominate over non-negligible Umklapp scattering events. Due to the noise limit of our measurements, this is not clear in Fig. 4c where α is varied between -1 and 0. We do not expect to reach the Poisson regime at lower temperatures where α is positive (below 100 K) and even less the Casimir regime, at low temperatures, where phonons are considered ballistic and the mean free path is of the same order of the sample lenght, i.e., 8 μm in our case. Notably, the heat transfer was ballistic in suspended nanowire[40] with a similar approach. It results in a different behavior compared to results in Fig. 4.

However, it is important to discuss the observed nonlinear temperature profile with respect to these different thermal transport regimes without considering the temperature range. In the ballistic regime, a phonon moves along the 2D material without interactions in both directions, and the temperature remains constant. A temperature drop occurs at the contact point, which is symmetric to the two contact points due to symmetry and convention[40]. In this coherent regime,

effect of resonances and oscillations can appear (e.g. Max-Zhender interferometer, Fabry-Perot cavity…). This raises many questions. The most important one is whether it is possible to measure the temperature of a ballistic phonon without interactions with Raman spectroscopy, in this case, where the local temperature measurement is more related to a local thermal bath of phonons. Raman thermometry assumes local thermal equilibrium between optical and acoustic phonons via mainly the thermoelastic effect. In our configuration, this condition is satisfied, as with the calibration itself (see Supplementary Information Section 3) and as demonstrated by the identical temperature evolution of WSe$_2$ and hBN Raman peaks, confirming strong interlayer thermal coupling. The method remains valid in the diffusive and hydrodynamic regimes but may become limited in the ballistic regime, which is expected to occur only below ~100 K in 2D materials.

A second question concerns the existence of oscillating thermal transport in such a system[53].

Between the classical phonon transport regime at high temperatures and the low-temperature regimes, the hydrodynamic regime dominates. Nonlinear behaviors could result in local variations in thermal conductivity. This empirical approach fits well with the hydrodynamic regime framework, even if the experimental observations remain to be defined quantitatively. The hydrodynamic regime was introduced in the 2D by solving the Peierls Bolzmann equation with the Callaway approximation[11,54] One approach to account for normal phonon interactions is to integrate the group displacements, with a group velocity of the phonons within the Brillouin zone. This model was not sufficient to describe the contribution of the collective effect of phonons in the 2D[12], but it describes the second sound of phonons in 2D material[11]. Again, it is difficult to quantitatively evaluate this notion in our sample, but it must introduce the interference of the wave packets at the interfaces and static oscillations of temperature along the sample.

We have developed a state-of-the-art experimental platform that combines confocal Raman thermometry with microheating-based thermal control to study the thermal conductivity of suspended 2D materials, where heat transport is inherently localized. This approach was made possible by an advanced 2D material transfer technique that enables the precise placement of 2D materials on suspended microheaters. Using a rigorous temperature calibration, we demonstrated that suspended monoisotopic h$^{10}$BN[55] exhibits a remarkably high in-plane thermal conductivity $k_{2D}$ of 1650 ± 550 W.m$^{-1}$.K$^{-1}$ at room temperature, exceeding previously reported values. This exceptional thermal performance is directly related to the large suspended sample geometry and isotopic purity, which reduces phonon scattering. Beyond the conventional two- and six-point measurements, we adopt a multipoint temperature mapping approach that has revealed fundamental deviations from the classical Fourier transport regime. The nonlinearities in the temperature profile of the suspended hBN are striking, a signature of an unconventional phonon transport mechanism. These results are obtained across different measurement conditions and open new avenues for investigating exotic thermal transport phenomena, such as the hydrodynamic regime in 2D materials. Our results support the use of 2D material as the next generation of thermal logic components, analogous to their electronic counterparts, such as thermal rectifiers, diodes and logic gates, where non-trivial thermal transport can play a fundamental role. To accelerate this transition, further progress in both theoretical modeling and experimental investigation of thermal transport at the nanoscale is essential. These advances are a crucial step towards using heat as an active degree of freedom in future device architectures.

## Methods
### Set-up
All measurements are performed in a modified Linkam cell under vacuum (10$^{-4}$ mbar) and at a low temperature achieved by liquid nitrogen cooling. The optical window of the Linkam cell was used to perform all Raman spectroscopy measurements using high resolution spectrometer (HR evolution spectrometer, 532 nm laser, Horiba). In our standard experimental procedure, heat was generated by Joule heating one of the microheater, establishing a stable temperature gradient across the suspended membrane. The temperature gradient over the 9 µm wide suspended region exceeds 30 °C/µm, ensuring precise thermal characterization.

### Sample fabrication
The suspended 2D material consists of either thin graphite flakes or a vertical WSe$_2$/hBN heterostructure. The monoisotopic h$^{10}$BN crystalline flake was grown by precipitation from a molten metal solution. The thicknesses of all materials (WSe$_2$, hBN, graphite) are confirmed by Raman spectroscopy and atomic force microscopy prior to membrane fabrication. The hBN and graphite materials are mechanically exfoliated layers of pure bulk crystals. The WSe$_2$ is obtained from a uniform monocrystalline WSe$_2$ flake (1 L to 3 L thick) grown on SiO$_2$/Si by chemical vapor deposition. The WSe$_2$ is picked up using a water droplet with a polypropylene carbonate film (PC) and stacked vertically over the hBN layer. The resulting WSe$_2$/hBN vdW heterostructure was cleaned with acetone and annealed at 300 °C in a low-pressure oven. The vertical WSe$_2$/hBN 2D vdW-HS was picked up again to be suspended across the gap between the two silicon cantilevers.

### Raman calibration
Several calibrations have been be carried out before actual measurements. In particular, the laser power must be carefully tuned using calibration experiments to eliminate any heating effects caused by the laser itself. In Raman thermometry, we track a specific Raman line frequency, $\omega(T)$, as a function of temperature, $T$. In a first approximation, $\omega(T)$ can be expressed as $\omega(T) = \omega_0 + \chi T$. The reference temperature is that of the ambient environment ($T_O = 293$ K) and during calibration the whole Linkam cell is heated uniformly and its temperature is measured via a PT100 sensor soldered on the sample holder. In this way we calibrate the Raman thermometry $\omega(T)$ and adjust the laser power to avoid any laser-induced heating. To complete the calibration, an additional measurement is performed to determine the temperature from the ratio between the Stokes and anti-Stokes intensities of the Si Raman LO peaks, which provides another independent temperature value. From these results we estimate the Raman thermal coefficients $\chi$ of -0.0106 cm$^{-1}$.K$^{-1}$ for WSe$_2$, -0.0192 cm$^{-1}$.K$^{-1}$ for Si, -0.0105 cm$^{-1}$.K$^{-1}$ for graphite and -0.0157 cm$^{-1}$.K$^{-1}$ for h$^{10}$BN.

### Optical strain and temperature separation in a vdW HS
In our 2D vdW HS, the thickness ratio of the monoisotopic h$^{10}$BN to WSe$_2$ t$_{hBN}$/t$_{WSe2}$ is more than 100 and the in-plane thermal conductivity ratio k$_{hBN}$/K$_{WSe2}$ is at least 20. Under stationary conditions, we can therefore assume that the heat flow in the 2D vdW HS is mainly via the isotopic hBN. This means that at each location of the membrane, the main heat source for the WSe$_2$ monolayer (0.6 nm thick) is the much thicker hBN support layer. The same is true for the mechanical stress in the 2D vdW HS, which is dominated by the response of the hBN material due to its much highest conductivity, relative thickness, Young Modulus and bending stiffness compare to WSe2. We can assume here that any compression, stretching or bending of the hBN is fully transferred to WSe$_2$, and we neglect any mechanical sliding between hBN and WSe$_2$ in the 2D vdW HS.

Based on these assumptions, we consider that the WSe$_2$ is subjected to the same strain and temperature than the underlying hBN. To validate this, we analyze the experimental variations of the $A_{1g}$ and $E_{2g}$ peaks of WSe$_2$ and hBN, respectively as shown in Fig. 1e. During this experiment, the 2D vdW HS was subjected to a controlled heat flow from the hot to the cold microheaters. The data

points follow two different trends representing the regions of the 2D vdW HS that are mainly affected by either mechanical deformation or thermal heating.

In the mechanically deformation zone, the relative frequency shifts of the hBN and WSe$_2$ lines were strictly governed by strain-induced compression. Assuming a small strain, $\varepsilon$, in linear region, the points align on a characteristic slope determined by the ratio of mechanical Raman shift parameter $\chi m$, with $\omega(\varepsilon) = \omega_0 + \chi m \varepsilon$. This slope can be estimated from tabulated data in literature, represented as black line, and we find that the corresponding zone in our diagram corresponds to a total strain variation of about 0.1% This quantifies the residual strain in the 2D vdW HS, which remains minimal due to the suspended geometry. Such absence of strong deformations is further confirmed by additional AFM and optical measurements, see Supplementary Information.

In the zone mainly subjected to temperature variation, the relative frequency shifts of the hBN and WSe$_2$ lines follow a linear trend governed by temperature variations. The corresponding points align on another line, with $\omega(T) = \omega_0 + \chi T$. The corresponding black lines are drawn in Fig. 1e.

Overall, there is good general agreement between the Raman data and this simple model. For the selected Raman lines and materials, strain has a strong effect on the hBN peak position, while temperature is the main contributor to the WSe$_2$ peak position.

## Data availability

The data that support the findings of this study are available from the corresponding author. They are no restrictions to accessing data. The Comsol simulation are provided in the Supplementary Information section. Source data are provided with this paper.

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

## Acknowledgements

We thank Anis Chiout, Jérôme Saint-Martin and Michele Lazzeri for fruitfull discussion. The work was supported, by French grants ANR ANETHUM (ANR-19-CE24-0021, J.C.), ANR Deus-nano (ANR-19-CE42-0005, J.C.), ANR 2DHeco (ANR-20-CE05-0045, J.C.), ANR Comodes (ANR-22-CE09-0021, J.C.)), ANR ELEPHANT (ANR-21-CE30-0012-01, J.C.), and (ANR-22-PEXD-0006, J.C.) FastNano project, as well as the French technological network RENATECH, J.C. Support for the monoisotopic hBN crystal growth and was provided by the USA Office f Naval Research award N00014-22-1-2582 (J.H.E.).

## Author contributions

J.C. and A.O. initiated the work. C.B.-R. fabricated the 2D hetero-structures, developed the soft 2D transfer and did the measurements with calibration. C.W. and F.P. fabricated the microheater. T.P. and J.H.E. have grown the isotopic hBN samples with the help of B.G. and G.C. G.D.B. and F.O. have grown the CVD WSe2 flakes. M.L.D. proceeded to the thermal reflectance measurements. C.B.-R., S.S., N.B., and L.M. are responsible for the graphene sample and measurements. C.B.-R., J.C., and E.H. did the PDMS 2D stamp and sample preparation. J.C. guided the research and wrote the manuscript with the input from all the authors.

## Competing interests

The authors declare no competing interests.
