## [Transparent Peer Review file · Nature Communications]

Extreme Longitudinal Thermal Conductivity and Non-Diffusive Heat Transport in Isotopic hBN

Corresponding Author: Dr Julien Chaste

Version 0:

Reviewer comments:

Reviewer #1

(Remarks to the Author)

In this paper, the authors established a thermal characterization platform for suspended 2D materials. The setup consists of a 2D material sample thermally actuated by silicon microheater thermal bridges, with the temperature profile measured by Raman shift temperature mapping. The 2D material thermal conductivity can be derived through fitting towards a finite element heat transfer model that resembles the actual device. The authors then demonstrated the setup through measuring thermal conductivity of monoisotopic hBN material, and identified unique non-linear heat transfer effects happening under specific temperature ranges. Overall, the authors developed an interesting technical pathway towards 2D material thermal characterization, and recorded some unconventional phonon transport phenomena in 2D hBN. However, the motivation and novelty of this work may deserve better clarification, and the experimental results may be discussed in a more logical and rigorous way. I would recommend the manuscript for publication if the following questions were addressed.

1. The contribution and novelty of the current work may be enhanced in the introduction part. Although the authors reviewed the thermal measurement of 2D materials and clarified the uniqueness of monoisotopic hBN, the thermal conductivity has already been measured elsewhere. Also, Raman temperature mapping itself is already a commercialized technology. How the proposed thermal characterization platform contributes to the field may need some clarification.
2. The experimental procedure and results may be narrated in more logical way to prevent confusion. In the 2-point and 4-point thermal conductivity measurements, a valid logic may be: at which spots are Raman shift temperatures measured, how the results look like, and how the simulation is fitted towards the measured temperature to calculate k_{2D} . For example, figure 3 c reported 13 temperature sampling points, but only 6 temperatures are used in the fitting. What are the 6 used temperatures, why is it relevant to measure 13 temperature points, and why not directly fitting the 13 temperatures in figure 3 c towards the simulation model?
3. What does the term 'LMM' refer to, and what are the differences among the three samples LMM1, LMM2, and LMM3? In figure 2 there is a caption of LLMGr, is LLM same as LMM?
4. In Figure 3 c, a very low temperature variation is reported around the middle of the 2D material, while the uncertainty becomes much larger at two sides. Why is there a large difference of uncertainty in figure 3 c? For example, at $x = 7 \text{ um}$ the uncertainty is extremely large. Does it mean a poor adhesion of 2D material on Si bridges?
5. In figure 3 d, the authors compare the measured hBN conductivity to literature. However, both datapoints are larger than the literature reported result. As no special treatment is explicitly reported in the paper as improvement of thermal conductivity, readers may expect same numbers with literature as a proof of measurement accuracy. What caused this measurement error compared to literature?
6. Why are two thermal conductivities reported in figure 3 d? Are they from the same sample of LMM1? Why are they about 50% different in values?
7. In figure 4 a, how is the $k \sim T^{-1}$ relationship derived? It seems more like a linear temperature distribution with $k \sim T^0$.

8. Figure 4 e shows a reversed temperature bias around the cold end, and it seems to be a violation of thermodynamic second law from the first glance. More clarifications may be necessary here.

9. In figure 5 a, the forward biased and reverse biased samples seem to have different lengths, as the black dots are not aligned for the two curves. Is this a plot issue, or are there specific reasons for this?

10. The non-linearity of temperature in figure 4 and the parabolic temperature distribution in figure 5 may deserve better clarification. It is understandable that an explanatory theoretical framework may be hard to derive. But can the results be explained qualitatively by some extraordinary phonon transportation mechanisms? Or are there any previous works reporting similar phenomena?

Reviewer #2

(Remarks to the Author)

The authors investigate suspended monoisotopic hBN/WSe₂ heterostructures using joule heating and Raman thermometry. They report exceptionally high in-plane thermal conductivities (up to 1650 W/m.K) and interpret non-linear spatial profiles as evidence for non-diffusive (hydrodynamic/ballistic) phonon transport. The topic is timely, and the experimental approach is interesting. The experimental platform, which combines microheaters and spatially resolved Raman mapping, is innovative, and the attempt to probe non-diffusive phonon transport is ambitious. However, the conclusions are not fully supported by the evidence provided because of the following reasons:

1- Raman thermometry assumes local thermal equilibrium. Under steep gradients or in ballistic/hydrodynamic regimes, the probed phonon populations may not be thermalized, making Raman shifts an uncertain indicator of temperature. The manuscript acknowledges this conceptual issue but does not validate Raman as a thermometer under these conditions.

2- The assumption that WSe₂ Raman modes track hBN temperature is only calibrated under uniform heating. No data are shown to confirm equivalence in high-gradient, nonequilibrium regimes.

3- Potential probe heating, residual strain, and interfacial resistance could significantly affect extracted values. The COMSOL fitting involves strongly correlated parameters, but no systematic sensitivity or uncertainty analysis is presented.

4- The conclusion that nonlinear profiles indicate hydrodynamic transport is premature. Diffusive models with temperature-dependent conductivity, inhomogeneous heating, or contact variation could also explain the observations.

5- Claims of "highest conductivity" and "robust evidence" are not justified given the uncertainties and lack of validation. Perhaps presenting anti-Stokes/Stokes temperature maps alongside shift-derived results would help substantiate the conclusions. The manuscript would also be more convincing if the authors could demonstrate that the Raman probe laser does not introduce measurable heating. Cross-validation with an independent thermometry technique would further strengthen the work. In addition, a full parameter sensitivity analysis of the COMSOL fitting would be valuable. Finally, the authors are encouraged to quantitatively assess whether diffusive models, incorporating realistic parameter variations, could reproduce the observed profiles.

Overall, the manuscript presents promising methods and intriguing data, but currently over-interprets results. Major revision is required to establish the validity of Raman thermometry in non-equilibrium regimes and to exclude diffusive explanations.

Version 1:

Reviewer comments:

Reviewer #1

(Remarks to the Author)

The authors' detailed response letter addressing the review comments is greatly appreciated. The revised manuscript indicates a significant improvement on clarification of novelty, and a better narration on the experimental results and discussions. Overall, I recommend the manuscript for publication, subject to the following minor revisions.

1. In the discussion of Figure 3, the authors clarified the 6-point fitting strategy by mentioning "increasing the number of sampling points does not improve the fitting quality". It would be good if evidence (maybe a figure comparing different fitting strategies) could be provided in the supplementary material to support this.

2. The sample nomenclature (HS1, HS2, Gr1) is updated in the main text, but not in the supplementary information. It would be good to make them unified.

Reviewer #2

(Remarks to the Author)

The revised manuscript addresses most of my concerns in a satisfactory manner. The authors have clarified the experimental methodology, improved the presentation of the multi-point thermal measurements, and significantly moderated their claims regarding non-diffusive heat transport. The exceptionally high thermal conductivity values are now better contextualized through control experiments and a clearer discussion of sample geometry, isotopic purity, and suspension

effects. While some interpretations, particularly regarding the validity of Raman thermometry under strongly non-diffusive conditions and the precise physical origin of the observed non-linear temperature profiles, remain suggestive rather than conclusive, the authors now appropriately acknowledge these limitations and avoid overstatement. Overall, the work presents a technically innovative experimental platform and compelling phenomenological observations that will be of broad interest to the thermal transport and 2D materials communities. I therefore recommend publication after minor revision, provided that the cautious framing of the conclusions is maintained throughout the manuscript.

Version 2:

Reviewer comments:

Reviewer #1

(Remarks to the Author)

After carefully reading the response letter and revised manuscript, I believe the authors have addressed all my concerns. I would recommend the manuscript to be accepted by Nature Communications.

Reviewer #2

(Remarks to the Author)

The authors have adequately addressed all points raised by the reviewers. I therefore recommend that the manuscript be accepted for publication in Nature Communications.

Extreme Longitudinal Thermal Conductivity and Non-Diffusive Heat Transport in Isotopic hBN" to Nature Communications.

REVIEWER COMMENTS

Reviewer #1 (Remarks to the Author):

In this paper, the authors established a thermal characterization platform for suspended 2D materials. The setup consists of a 2D material sample thermally actuated by silicon microheater thermal bridges, with the temperature profile measured by Raman shift temperature mapping. The 2D material thermal conductivity can be derived through fitting towards a finite element heat transfer model that resembles the actual device. The authors then demonstrated the setup through measuring thermal conductivity of monoisotopic hBN material, and identified unique non-linear heat transfer effects happening under specific temperature ranges. Overall, the authors developed an interesting technical pathway towards 2D material thermal characterization, and recorded some unconventional phonon transport phenomena in 2D hBN. However, the motivation and novelty of this work may deserve better clarification, and the experimental results may be discussed in a more logical and rigorous way. I would recommend the manuscript for publication if the following questions were addressed.

1. The contribution and novelty of the current work may be enhanced in the introduction part. Although the authors reviewed the thermal measurement of 2D materials and clarified the uniqueness of monoisotopic hBN, the thermal conductivity has already been measured elsewhere. Also, Raman temperature mapping itself is already a commercialized technology. How the proposed thermal characterization platform contributes to the field may need some clarification.

Answer: We appreciate the reviewer's insightful comment. We agree that both Raman thermometry and thermal conductivity measurements of hBN have been widely reported. However, our work introduces a distinct experimental platform that overcomes key limitations of conventional optical thermometry. In most Raman-based studies, the heat flux is induced by laser absorption, requiring precise knowledge of the optical absorption coefficient of the 2D material. This indirect heating approach often introduces large uncertainties and restricts quantitative comparison between samples.

In contrast, our setup employs custom-designed microheaters that provide a well-calibrated and spatially uniform heat flow through Joule heating, while temperature is mapped optically with submicron resolution. This combination enables the direct measurement of temperature profiles along suspended monoisotopic hBN heterostructures, under controlled and reproducible thermal bias. This is a capability that is rarely demonstrated in previous studies.

This approach allows not only the extraction of accurate in-plane thermal conductivity values but also the identification of non-Fourier heat transport regimes in monoisotopic hBN. The platform thus bridges the gap between purely optical thermometry and complex nanofabricated heater-sensor systems, providing a versatile and quantitative method for studying phonon hydrodynamics in 2D materials.

Here is a new paragraph in the introduction and we also modified the introduction in accordance to the comment and we reduce the length:

“Despite these advances, direct and quantitative mapping of thermal transport in 2D materials remains highly challenging. Most Raman-based measurements rely on optical heating, which requires precise knowledge of the absorption coefficient and often leads to large uncertainties in the extracted conductivity values. Furthermore, very few studies have directly measured spatial temperature profiles along suspended samples subjected to a controlled thermal gradient.

In this work, we introduce a custom-built thermal characterization platform that combines Joule-heated microheaters, providing a well-defined and tunable heat flux, with spatially resolved Raman thermometry. This approach enables direct optical mapping of temperature distributions under fully calibrated heating conditions. It thus provides a unique capability to quantitatively probe non-Fourier and hydrodynamic phonon transport in monoisotopic hBN or other 2D materials.”

“Raman temperature mapping itself is already a commercialized technology” Raman mapping is indeed a commercial tool, especially in our case but Raman temperature mapping is not, as far as we know. It necessitate a precise calibration that company can have difficulties to afford at the moment. It would

be great for us. We are in collaboration with Horiba France for the Raman (basically, there are our neighbour at Palaiseau and we detain a prototype of their company for SHG) and they have commercial Raman mapping but nothing as Raman temperature mapping. We did not heard about Renishaw or Witec or Edinburgh instruments solutions neither. We know, through a project, that Parks system wants to develop a commercial SThM, which is quite interesting as well.

2. The experimental procedure and results may be narrated in more logical way to prevent confusion. In the 2-point and 4-point thermal conductivity measurements, a valid logic may be: at which spots are Raman shift temperatures measured, how the results look like, and how the simulation is fitted towards the measured temperature to calculate k_{2D} .

Answer: Since this question is raised, we understand the need to reinforce the description of the 2- and 6-point measurements. One of the important messages of the article is that, in many cases, the minimum number of points to be measured is only 2 or 6, but this is not automatic. In general, we systematically performed temperature mapping with about 40 points for almost all samples. The real purpose of the discussion (and our reasoning) is to determine the minimum number of points necessary for a meaningful simulation, considered to be a good approximation, for each case.

For a simple experimental situation, two points are sufficient and we set the graphene framework. For the measurement of the heterostructure, a minimum of six points are required to properly model the temperature profile and we provide the framework (classical regime). To be on the safe side, we check each time that all the measurement points are correctly modeled by extrapolating the simulation and the classical Fourier law. In the context of Figure 4, we realize that all points must be considered to describe the heat transport regime. Therefore, we set limits and a framework for conductivity measurements in 2D materials for future experiments, which are often limited to a two-point measurement.

To address this comment, the introduction have also been expanded to highlight how this methodological clarification can guide future thermal measurements on 2D materials.

“The comparison between 2-point and 6-point measurements highlights the minimum conditions required for a reliable modeling of heat transport in 2D materials. This framework provides practical guidance for determining when a simplified approach is valid and when a full temperature mapping is necessary. The methodology can thus serve as a reference for future thermal conductivity studies in graphene and related heterostructures.”

And in the abstract **“This study establishes a practical framework linking Raman-based temperature mapping, the number of measurement points, and thermal simulations to reliably determine the in-plane thermal conductivity of 2D materials.”**

For example, figure 3 c reported 13 temperature sampling points, but only 6 temperatures are used in the fitting. What are the 6 used temperatures, why is it relevant to measure 13 temperature points, and why not directly fitting the 13 temperatures in figure 3 c towards the simulation model?

Answer: Indeed, the fitting shown in Figure 3c (and the simulated temperature map in Figure 3b) was performed in COMSOL using only six measurement points, corresponding to the six circles displayed in Figure 3a. We initially tested the model with a higher number of data points (13 points as shown in Figure 3c, and over 100 points in the full mapping of Figure 3b), but the resulting fits did not provide additional information or improve the accuracy of the extracted thermal conductivity. This confirms that, within the classical thermal transport regime, a limited number of temperature points (typically six) is sufficient to reconstruct the full thermal profile with high fidelity.

We acknowledge that this aspect was not clearly stated in the manuscript, and we have clarified it in the revised version.

We modified the figure 3 to emphasized the 6 point of measurements

And Figure 2 To clarify where the two pint of temperature measurement have been done

We replace the corresponding line in the text in order to add informations:” As verified through full temperature mapping, Figure 3a,3b and 3c, only six measurement points are sufficient to accurately reproduce, with COMSOL, the experimental temperature profile across the entire sample when the system operates within the classical heat transport regime. Increasing the number of sampling points does not significantly improve the fit quality or the extracted thermal conductivity. This finding provides a practical guideline for optimizing Raman-based thermal measurements, emphasizing the balance between experimental efficiency and physical accuracy. All details about the extraction of thermal conductivity, from these temperature are presented in

supplementary information, section S8 to S11. Basically, we minimized the parameter $\Sigma \Delta T^2 = \sum_{points} (T_{mes} - T_{sim})^2$ with simulated T_{sim} and measured temperature $T_{measured}$ and a strict conditions for each point with $(|T_{mes} - T_{sim}| < \delta T)$.”

In addition, we add in the supplementary material, section 14, an example of the different spectra that have been obtained during the measurement of the Figure 4a and for the six points measurement.

14 Example of spectra during the measurements (Figure 4a)

Figure S14-28 The different Raman spectra reported for the sample HS2 for the different materials along the measurement of Figure 4a. The points used for the thermal conductivity are average values obtained at the area localized by the different point in the inset

3. What does the term ‘LMM’ refer to, and what are the differences among the three samples LMM1, LMM2, and LMM3? In figure 2 there is a caption of LLMGr, is LLM same as LMM?

Answer: LMM refer to a nomenclature (within the group at C2N) referring to the shape of the microheaters and of the heterostructure (we have tested many shapes before to converge to the shape presented in figure 1). In order to clarify this point for an external lecturer, we have modified the sample name in all the manuscript and SI (LLMGr become **Gr1**, LMM1 becomes **HS1**, LMM2 becomes **HS2**) HS means heterostructure here and Gr1 refer to graphene sample. We hope it will simplify the text.

4. In Figure 3 c, a very low temperature variation is reported around the middle of the 2D material, while the uncertainty becomes much larger at two sides. Why is there a large difference of uncertainty in figure 3 c? For example, at $x = 7 \mu\text{m}$ the uncertainty is extremely large. Does it mean a poor adhesion of 2D material on Si bridges?

Answer: In Figure 3c, each point along the x-direction represents the average of five measurement points along the y-direction, and the corresponding uncertainty reflects the standard deviation among these five points. It is true that the uncertainty is minimal near $x = 0 \mu\text{m}$ and becomes significantly larger near $x = 7 \mu\text{m}$. The reviewer is correct that this could indicate a physical effect, such as partial detachment or reduced adhesion of the 2D layer on the Si bridges. Although we had not initially considered this interpretation, it is plausible, even if the thermal contact are good.

But, another explanation is also relevant: the Raman peak intensity of WSe_2 is lower on the supported regions than on the suspended ones, the signal is reduced in those areas (certainly due to difference in the optical reflectance), which leads to a higher relative uncertainty. It is not necessarily an additional noise or an adhesion noise but a lower signal on noise ratio.

We add this Figure to confirm this point in section 14 of the supplementary information. It can be seen that the Raman peak of WSe_2 which are at the contact position are 3 times lower in amplitude than the peaks at the suspended region, which induced more noise for the same integration time, in this case.

We add in the main text. **The uncertainty on the WSe_2 temperature is higher in the supported region due to a lower Raman intensity (supp. Information S14).**

5. In figure 3 d, the authors compare the measured hBN conductivity to literature. However, both datapoints are larger than the literature reported result. As no special treatment is explicitly reported in the paper as improvement of thermal conductivity, readers may expect same numbers with literature as a proof of measurement accuracy. What caused this measurement error compared to literature?

Answer: We agree with the referee that it is not clear in the main text but we are sure of this result and we even expected this one. Our measurement on graphene confirms it is not a measurement error. Previous reported value are even outside the uncertainty of our measurement at $1650 \pm \sim 400 \text{ W/m/K}$. In the main text, we have the following sentence but the message is unclear and limited “As shown in Figure 3d, our value for k_{2D} of about $1650 \text{ W} \cdot \text{m}^{-1} \cdot \text{K}^{-1}$ represents the highest reported to date for hBN. As can be observed from the graph and explained in the Supplementary Information, the origin can be attributed to the multiple factors: the material under investigation is pure monoisotopic h10BN,¹ the sample is suspended, the dimensions of the sample are quite large, and the temperature is slightly below room temperature.”

We modified the corresponding paragraph in the main text “Recent studies have shown that the thermal conductivity of van der Waals materials strongly depends on whether the samples are supported or suspended, their isotopic purity, thickness, and the measurement temperature, see supplementary information section 12. Suspended samples generally exhibit conductivities two to three times higher than supported ones (figure 2c), while isotopically enriched hBN and graphene show up to a twofold increase compared to natural materials¹. Measurement over larger flakes improve the thermal conductivity within the context of the hydrodynamic regime in 2D material.² Thicker flakes also tend to display higher conductivities,³ contrary to earlier assumptions for monolayers, it is still a debate. In our measurements, the exceptionally high value of k_{2D} , in HS1, even compared to HS2, is consistent with these trends, reflecting a favorable suspended configuration and high material quality.”

6. Why are two thermal conductivities reported in figure 3 d?

Answer, there are the thermal conductivity measured for sample HS1(LMM1) and HS2 (LMM2) where the geometry and the temperature variation are different

We added the sample name in the Figure 3d for clarity

Are they from the same sample of LMM1?

Answer: No one point is for HS1 and the other for HS2

Why are they about 50% different in values?

Answer: This is a very good point, since we can only answer partially.

We know that both sample do not have the same length and same thickness and we consider that it imply, by definition that the thermal conductivity is not the same. And the main culprit is the hydrodynamic regime of phonon transport

For the length, we are sure about this since it is in accordance with previous result and theory (graphic below). Note that the thermal conductivity still increased when the length is 1 mm!

[Figure Redacted]

Lenght (μm)

Fugallo, G., Cepellotti, A., Paulatto, L., Lazzeri, M., Marzari, N., & Mauri, F. (2014). Thermal conductivity of graphene and graphite: collective excitations and mean free paths. *Nano letters*, 14(11), 6109-6114..

For the thickness, there is a dependance but it is much more complicated. It can increase when the thickness decrease (case of graphene but it is still not clear) and it can decrease when the thickness decrease (case of MoS2, see below) A lot of work has to be done to understand this.

[Figure Redacted]

Xiao, P., El Sachat, A., Angel, E. C., Ng, R. C., Nikoulis, G., Kioseoglou, J., ... & Sledzinska, M. (2024). MoS2 phononic crystals for advanced thermal management. *Science advances*, 10(13), eadm8825.

We modified the corresponding paragraphe in the main text **“Recent studies have shown that the thermal conductivity of van der Waals materials strongly depends on whether the samples are supported or suspended, their isotopic purity, thickness, and the measurement temperature, see supplementary information section 12. Suspended samples generally exhibit conductivities two to three times higher than supported ones (figure 2c), while isotopically enriched hBN and graphene show up to a twofold increase compared to natural materials¹. Measurement over larger flakes improve the thermal conductivity within the context of the hydrodynamic regime in 2D material.² Thicker flakes also tend to display higher conductivities,³ contrary to earlier assumptions for monolayers, it is still a debate. In our measurements, the exceptionally high value of k_{2D} , in HS1, even compared to HS2, is consistent with these trends, reflecting a favorable suspended configuration and high material quality.”**

7. In figure 4 a, how is the $k \sim T^{-1}$ relationship derived? It seems more like a liner temperature distribution with $k \sim T^0$.

Answer: we indeed write “The thermal gradient across the suspended heterostructures is quasi-linear, akin to the description of the Fourier regime, and exhibits a temperature dependence of k_{2D} as $k_{2D}(T) =$

CT^α , where C and α are fitting coefficients. The exponent α is close to -1 which allows a classical treatment of the data in the following analysis analogous to HS1.”

We agree it is quite difficult to differentiate between 0 and 1 for α , due to the noise and it is maybe closer to 0, which means a constant conductivity with temperature and a linear thermal gradient.

However, both possibility are in accordance with the classical description of thermal transport and it does not interact with the rest of the discussion (Figure S9, below).

In Figure 4a, we wrote: “The thermal gradient across the suspended heterostructures is quasi-linear, akin to the description of the Fourier regime, and exhibits a temperature dependence of k_{2D} as $k_{2D}(T) = CT^\alpha$, where C and α are fitting coefficients. The exponent α is close to -1 which allows a classical treatment of the data in the following analysis analogous to HS1.”

The fitted exponent α is more 0 than -1, as suggested by the referre, but both solution are suggesting a classical regime in which thermal conductivity decreases with temperature. However, we acknowledge that the uncertainty in the data makes it difficult to clearly distinguish between $\alpha = 0$ (constant k) and $\alpha = -1$ (inverse dependence). Both cases are consistent with a Fourier-type heat transport regime, and we check that this ambiguity does not affect the validity of the subsequent analysis (see Figure S9-22 below, pay attention that thermal conductivity, for $\alpha=-1$ is described in Figure S7-20 and $k_{2D} = 1180 \pm 150 W \cdot m^{-1} K^{-1}$) mentionned in figure 3d corresponds also to the case $\alpha = 0$.

In accordance with this comment, we modify the previous sentence in the main text as:

“The thermal gradient across the suspended heterostructures is quasi-linear, akin to the description of the Fourier regime, and exhibits a temperature dependence of k_{2D} as $k_{2D}(T) = CT^\alpha$, where C and α are fitting coefficients. The exponent α is close to 0, between 0 and -1 which allows a classical treatment of the data in the following analysis analogous to HS1. Both dependance $\alpha=-1$ and $\alpha=0$ have been compared by simulation and it does not affect the extracted thermal conductivity (See supplementary information S9)”

And the SI by adding this comment and dedicate the part S9 in the SI as in the following text

1. Simulations for HS2: temperature dependence of k_{2D} .

The temperature gradient measured along the HS2 at a voltage of 8 V is approximately 250 Kelvin. The thermal conductivity $k_{2D}(T)$ varies considerably within this range. This dependence can be derived directly from the temperature profile. Considering a bar of a material of length L, between two temperatures, T_{hot} and T_{cold} , it is possible to define the temperature profile in accordance with Fourier's law. To obtain the temperature along the x-longitudinal axis, it is possible to introduce a temperature dependence such as $k_{2D}(T) = cT^\alpha$. The temperature profile is then written as follows ⁴ :

$$T(x) = \left[\frac{x}{L} T_{cold}^{\alpha+1} + \left(1 - \frac{x}{L} \right) T_{hot}^{\alpha+1} \right]^{\frac{1}{\alpha+1}}$$

In Figure 4c, the temperature profile of the sample LMM2 is quasi linear and corresponds to a coefficient α close to -1. It matches the expected behavior and experimental data obtained in reference ⁵.

A temperature dependence for $k_{2D}(T)$ was included in the COMSOL model. This model is based on the experimental values of $k_{2D,ref}(T)$ as reported in reference ⁵. Thus, $k_{2D}(T)$ is defined as:

$$k_{2D}(T) = k_{2D,ref}(T) \frac{k_{2D}(190K)}{k_{2D,ref}(190K)}$$

Nevertheless, the inclusion of the temperature dependence in k_{2D} has a minimal impact on the conductivity value of the 2D material at room temperature, as shown in Figure S9-22 when comparing the thermal conductivity with temperature dependence at 300K and the thermal conductivity without temperature dependence. We can compare at 300 K which corresponds to the mean of the vdW structure temperature.

Figure S9-22: Simulations for the sample HS2 and phase space of the converging solutions a) with (left) and b) without (right) temperature dependence of k_{2D} .

8. Figure 4 e shows a reversed temperature bias around the cold end, and it seems to be a violation of thermodynamic second law from the first glance. More clarifications may be necessary here.

Answer: We appreciate the reviewer’s insightful question. We do not claim any violation of the second law of thermodynamics, as we have no evidence supporting such a conclusion.

We have added the following clarification, in bold, to the Supplementary Information, section 15 with the additional graph:

“The apparent reversed temperature bias near the cold end in Figure 4e is indeed an intriguing observation that may warrant further investigation if confirmed. At this stage, however, we believe the effect remains within the bounds of experimental uncertainty and cannot be conclusively interpreted unless reproduced consistently. As shown in the expanded plot below, the deviation occurs over only a few data points around $x = 10 \mu\text{m}$, with an amplitude of roughly two standard deviations, suggesting a small signal close to the measurement noise rather than a genuine temperature inversion.”

Regarding the concern about a possible violation of the second law of thermodynamics, we offer the following points for clarification:

The system is not isolated; it is continuously driven out of equilibrium by Joule heating at one contact.

Heat transport in such mesoscopic structures involves both forward and backward phonon fluxes, analogous to the Landauer picture for electronic conduction. This highlights that the interpretation of heat flux can be more complex than the macroscopic formulation of the second law and requires considering detailed phonon interactions.

The apparent local reversal may simply reflect complex phonon population dynamics, involving diverse branches, energies, and momenta, rather than an actual decrease in entropy. In regimes approaching phonon hydrodynamics, simplified models such as the SMA become inadequate.

Related phenomena, including local phonon cooling or parametric squeezing, for example, the one that have been reported in graphene optomechanical systems (e.g., Barton *et al.*, *Nano Lett.* 12, 4681 (2012)) and other quantum systems. Raman processes themselves can pump or cool specific

phonon modes through Stokes and anti-Stokes interactions, illustrating how localized phonon cooling may occur.

Raman thermometry probes primarily the optical phonon population, which may transiently decouple from acoustic phonons in a nonlinear thermal landscape, making a strict thermodynamic interpretation difficult.

A comprehensive understanding of this effect will require dedicated low-temperature experiments and more advanced modeling. “

9. In figure 5 a, the forward biased and reverse biased samples seem to have different lengths, as the black dots are not aligned for the two curves. Is this a plot issue, or are there specific reasons for this?

We thank the reviewer for this comment. To clarify this point, we have revised Figure 5a by adding the silicon temperature at the position of the right microheater. This value is obtained with a higher uncertainty, as the spatial determination is less precise in this region. In the revised manuscript, we now state: “In Figure 5a, the temperature of the silicon microheater at 11 μm , for the reversed heat-flow case, was estimated with higher uncertainty and position due to missing data.” Below we detail the exact procedure followed for Figure 5a.

First, we replotted the original forward-bias data including all points

for which a temperature could be extracted from the Raman peak. This revealed a few additional points corresponding to measurements at the very edge of the silicon microheater. We also observed a small shift of a few tens of degrees near the microheater edge. These points were removed because the Raman intensity there is very low, leading to non-physical deviations in the peak fitting. At the edges, the Raman signal is weak due to convolution with the Gaussian laser profile. Since the microheater thickness is about $3\ \mu\text{m}$, the laser can probe the bottom of the silicon layer even slightly outside the nominal microheater region. At this depth, the laser is not perfectly focused and the lateral resolution degrades by several micrometers (as illustrated in the schematic). Consequently, determining the exact microheater edge using the silicon Raman peak is intrinsically difficult. For this reason, the silicon temperature was determined, for simulation, a few micrometers away from the edge, where the signal becomes more stable.

Regarding the reversed-bias configuration in Figure 5a, it was not possible to measure the temperature of the second microheater because the alignment with the device was not optimal during acquisition. We clearly miss some data but the measurement is took one day and we did not have time to launch it again. The black dot shown in the figure corresponds to a Raman signal of very low intensity, likely recorded outside the microheater region. In that case, an apparent deviation of several tens of degrees is expected due to the poor signal-to-noise ratio.

We change the Figure 5 as

10. The non-linearity of temperature in figure 4 and the parabolic temperature distribution in figure 5 may deserve better clarification. It is understandable that an explanatory theoretical framework may be hard to derive. But can the results be explained qualitatively by some extraordinary phonon transportation mechanisms? Or are there any previous works reporting similar phenomena?

Answer ; there are not previous report of similar measurement or theory for the Figure 4. The parabolic behavior in Figure 5 is expected theoretically in the hydrodynamic regime (Li, X., & Lee, S. (2018). Role of hydrodynamic viscosity on phonon transport in suspended graphene. *Physical Review B*, 97(9), 094309. And more recently So, S., Seol, J. H., & Lee, J. H. (2024). Quasiballistic thermal transport in submicron-scale graphene nanoribbons at room-temperature. *Nanoscale Advances*, 6(11), 2919-2927.). This reference is cited in the paper But is is complicate to fully translate the profile obtained in the model and our result. It has not been measure exepimentally so far.

[Figure Redacted]

We simply discuss about this with multiple theorician working on both 2D material and thermal transport (Michele Lazzeri, Georgia Fugallo, Konstantinos Termentzidis, Xavier Alvarez, Jerome Saint Martin), about our result and the conclusion of the discussions was simple; the present result are very interesting but it implies “more data” to performs the corresponding model. With a basic formula, it gives; any model can be adapted to fit a non linear curve if it has more than 6 parameters. It is clearly something that we will provide in the future. we even organized a “journée thématique”, (<https://howdi-name.sciencesconf.org/?lang=en>) in order to talk about this question). We will publish soon a second paper for similar measurements obtained with two laser, one for heating and one for reading the temperature. The circular configuration of this experiment is much more adapted to modelisation, all the details will be indicated in the publication with the theory which was done by Michele Lazzeri.

Reviewer #2 (Remarks to the Author):

The authors investigate suspended monoisotopic hBN/WSe₂ heterostructures using joule heating and Raman thermometry. They report exceptionally high in-plane thermal conductivities (up to 1650 W/m.K) and interpret non-linear spatial profiles as evidence for non-diffusive (hydrodynamic/ballistic) phonon transport. The topic is timely, and the experimental approach is interesting. The experimental platform, which combines microheaters and spatially resolved Raman mapping, is innovative, and the attempt to probe non-diffusive phonon transport is ambitious. However, the conclusions are not fully supported by the evidence provided because of the following reasons:

1- Raman thermometry assumes local thermal equilibrium. Under steep gradients or in ballistic/hydrodynamic regimes, the probed phonon populations may not be thermalized, making Raman shifts an uncertain indicator of temperature. The manuscript acknowledges this conceptual issue but does not validate Raman as a thermometer under these conditions.

Answer: We comment on the main text about this by saying: “This (non linearities) raises many questions. The most important one is whether it is possible to measure the temperature of a ballistic phonon without interactions with Raman spectroscopy, in this case where the local temperature measurement is more related to a local thermal bath of phonons.”

As mentioned in the manuscript, Raman thermometry indeed assumes local thermal equilibrium, and this assumption may not hold under strongly non classical conditions such as ballistic or exotic regime. In the classical regime or diffusive regime, as the hydrodynamic regime, however, Raman spectroscopy remains a reliable method for local temperature measurements in 2D material, in out of equilibrium measurements, since optical phonons are well coupled to the thermal bath of the crystal through frequent phonon–phonon interactions. This is validated in our study, as in most of the litterature, by the calibration of Raman peak positions versus temperature (see supplementary information section 3). It is usually necessary to consider the contributions from thermal expansion and phonon anharmonicity (including 3-ph and 4-ph scattering processes)(Liu, H. N., Cong, X., Lin, M. L., & Tan, P. H. (2019). The intrinsic

temperature-dependent Raman spectra of graphite in the temperature range from 4K to 1000K. Carbon, 152, 451-458.)).

The thermal bath of a crystal is generally associated with its population of acoustic phonons. In pump-probe experiments, acoustic phonons can be excited optically, either directly through the photoelastic effect or indirectly via laser-induced thermomodulation. (Ge, S., Liu, X., Qiao, X., Wang, Q., Xu, Z., Qiu, J., ... & Sun, D. (2014). Coherent longitudinal acoustic phonon approaching THz frequency in multilayer molybdenum disulphide. Scientific reports, 4(1), 5722.) The pump pulse produces a sudden temperature rise that generates strain through thermal expansion. In most cases, these mechanisms involve an initial coupling between light and optical phonons, which subsequently transfer energy to the acoustic phonons, establishing the thermal equilibrium within the lattice. This can highlight the “thermal relation” between optical phonons and acoustical phonons in a 2D material.

In our specific configuration, the temperature of hBN is probed indirectly (remotely) through the Raman signal of the adjacent WSe₂ layer, which acts as a local thermometer (see supplementary information section 4). The excellent agreement between the Raman shifts of WSe₂ and hBN (Figure 1e) confirms that the two layers are thermally well-coupled when submitted to a temperature gradient. Thus, Raman thermometry measure the local thermal bath temperature and remains valid for our system, which operates at ambient temperature and within the diffusive-to-hydrodynamic regime, at less with phonons interactions. We agree that in the purely ballistic regime, Raman thermometry becomes questionable, but such a regime is expected only at much lower temperatures (< 100 K in 2D materials) and this remains very delicate to measure, regardless of the method (see Tavakoli et al., Nat. Commun. 9, 4287 (2018); Jezouin et al., Science 342, 601 (2013)).

Modification of the main text, we add

Raman thermometry assumes local thermal equilibrium between optical and acoustic phonons via mainly thermoelastic effect. In our configuration, this condition is satisfied, as with the calibration itself (see supplementary information section 3) and as demonstrated by the identical temperature evolution of WSe₂ and hBN Raman peaks, confirming strong interlayer thermal coupling. The method remains valid in the diffusive and hydrodynamic regimes but may become limited in the ballistic regime, which is expected to occur only below ~100 K in 2D materials.

2- The assumption that WSe₂ Raman modes track hBN temperature is only calibrated under uniform heating. No data are shown to confirm equivalence in high-gradient, nonequilibrium regimes.

Answer: We thank the reviewer for this relevant remark since it is a very important point and Figure 1e is centrale in our argumentation. We clarify that Fig. 1e is obtained from the same out-of-equilibrium experiment shown in Fig. 5, where sample HS3 is subjected to a controlled thermal gradient between the microheaters. This measurement is therefore performed in a non-uniform, high-gradient regime. In this configuration, the Raman shifts of WSe₂ and hBN remain aligned along the temperature-calibration curve (black line), demonstrating that the WSe₂ Raman modes faithfully track the hBN temperature even under strong nonequilibrium conditions.

The manuscript has been modified accordingly. **“It is measured under an out-of-equilibrium conditions with a controlled thermal gradient across the device. The alignment of the WSe₂ and hBN Raman shifts along the calibration curve confirms that WSe₂ Raman modes reliably track the hBN temperature even under high-gradient, nonequilibrium regimes.”**

3- Potential probe heating, residual strain, and interfacial resistance could significantly affect extracted values. The COMSOL fitting involves strongly correlated parameters, but no systematic sensitivity or uncertainty analysis is presented.

Answer: We fully agree with the reviewer that probe-induced heating, residual strain, interfacial resistance, and the correlation between fitted parameters are critical aspects that must be assessed to ensure the robustness of the extracted thermal properties. To address these concerns, we have revised the manuscript to better connect the main text with the Supplementary Information (SI), where we present a systematic evaluation of these effects.

In the main text, we now clarify: “ **We demonstrate in the supplementary information, that the measured temperature profile is not affected by residual strain (S5), laser probe heating (S4) and we proceed to analyses and discussion, with uncertainty analyses, along the supplementary information for microheater and 2D thermal conductivities (S6, S8),, temperature dependance of the thermal conductivity (S9) the hBN out-of plane thermal conductivity (S10), interfacial thermal resistance (S11)**”

To further clarify how uncertainties are handled, we added:

“**All details about the extraction of thermal conductivity, from these temperature are presented in supplementary information, section S8 to S11. Basically, we minimized the parameter $\Sigma \Delta T^2 = \sum_{points} (T_{mes} - T_{sim})^2$ with simulated T_{sim} and measured temperature $T_{measured}$ and a strict conditions for each point with $(|T_{mes} - T_{sim}| < \delta T)$.**”

Regarding the sensitivity analysis, we acknowledge the reviewer’s remark and clarify the reason for not including a COMSOL-based sensitivity extraction in the initial version: although COMSOL provides tools for multi-parameter sensitivity analysis, we found that the results were not fully reliable in our case because: (i) the extraction required advanced configuration beyond our current expertise, (ii) The second is our concern to understand properly the physical meaning of our result through Comsol (we had to test the comsol simulation by many different means to be 100% sure of the simulations present along the manuscript) and (iii) the sensitivity interpretation was not straightforward for strongly correlated thermal parameters. For this reason, we instead performed a series of controlled parameter-variation studies presented in Sections S8–S11, which provide a transparent and physically interpretable assessment of parameter uncertainty and correlation.

In the literature (and in comsol), sensitivity is the partial derivative of a parameter in function of another. i.e. in our case $\partial T / \partial k$ for example with k the conductivity of the silicon, the conductivity of the 2D heterostructure, the interfacial resistance... In the reference (Yuan, Cet al. M. (2019). Modulating the thermal conductivity in hexagonal boron nitride via controlled boron isotope concentration. Communications physics, 2(1), 43.) They consider the time derivative $\partial T / \partial t$ since it is a time resolved experiment. It is also possible to find publication with the sensitivity defined by $\partial \ln(T) / \partial \ln(x)$. In our case, we can define S as $\partial T / \partial k$ at every position x along the sample and we apply it to the case of graphene in Figure 2.

In Figure 2a, $S(k_{Si}) \sim \frac{12 K}{20 W \cdot m^{-1} K^{-1}} = 0.6 W^{-1} \cdot m \cdot K^2$ for x= - 5 μm , $S(k_{Si}) = 0.5 W^{-1} \cdot m \cdot K^2$ for x=2 μm and $S(k_{Si}) = 0.5 W^{-1} \cdot m \cdot K^2$ for x= 8 μm respectively

In Figure 2a, $S(k_{Si}) \sim \frac{6 K}{1500 W \cdot m^{-1} K^{-1}} = 0.004 W^{-1} \cdot m \cdot K^2$ for x= - 5 μm , $S(k_{Si}) = 0 W^{-1} \cdot m \cdot K^2$ for x=2 μm and $S(k_{Si}) = 0.003 W^{-1} \cdot m \cdot K^2$ for x= 8 μm respectively.

We add in inset of Figure 2a and 2b these points for the sensitivity measurements in Figure 2a and 2b. and add in the caption “In inset the sensitivity S as $S = \partial T / \partial k$ ”

About the temperature dependance of the thermal conductivity (S9) the hBN out-of plane thermal conductivity (S10), we consider it is already very difficult to understand the meaning of sensitivity since we simulate the thermal contact and the temperature dependance of the conductivities with very specific methods and difficult to translate in sensitivity S (maybe by taking the coefficient α AND the conductivity at a certain temperature). We can also consider that the sensitivity to the out of plane conductivity of hBN is weak, since the effect is reduced (see S10).

The notion of sensitivity define in Comsol ($S = \partial T / \partial t$) indicates a relation with the noise of our system. If a parameter P has some noise, typically a random variation δP which induces a noise in temperature it transduces in temperature noise $\sqrt{\langle \delta T^2 \rangle} = \partial T / \partial P \sqrt{\langle \delta P^2 \rangle}$. And the sensitivity is defined when we measure a signal on noise ratio which is equal to Signal/Noise=1. The noise of temperature is related to the parameter $\Sigma \Delta T^2 = \sum_{points} (T_{mes} - T_{sim})^2$ defined in the article (with a simple normalisation due to the number of points). It is possible to relate the notion of sensitivity by looking at the variation of $\Sigma \Delta T^2$ in function of small variation of δP or P values. We preferred to used it along the supplementary information as it relate the definition describe above and the notion of sensitivity with partial derivative

Another point of view on sensitivity and uncertainty is graphic approach (see wikipedia : https://en.wikipedia.org/wiki/Sensitivity_analysis). In fact, we plot the graph $\Sigma \Delta T^2 (k_{Si}, k_{vdw})$ as done below to determine the sensitivity of our system and the uncertainty at the same time in a control manner (see Section S11 of supplementary information but also S8, S9, S10)

4- The conclusion that nonlinear profiles indicate hydrodynamic transport is premature. Diffusive models with temperature-dependent conductivity, inhomogeneous heating, or contact variation could also explain the observations.

Answer: The reviewer suggests that nonlinear temperature profiles could arise from diffusive models featuring temperature-dependent thermal conductivity, inhomogeneous heating, or contact variations. Below we explain why these mechanisms cannot account for the observed nonlinearities.

1. Temperature-dependent conductivity

For HS2, the nonlinear temperature profile shown in Figure 4 requires careful consideration of the sample geometry, particularly the suspended region. In Figure 5a, the temperature profile across the suspended part of the 2D heterostructure remains quasi-linear. We explicitly model thermal conductivity k_{2D} as $k_{2D}(T) = CT^\alpha$ and show profiles for various values of α . As shown in Fig. 5e, none of these temperature-dependent diffusive models produces a nonlinear temperature profile comparable to the experimental one. Thus, the observed nonlinearity cannot be attributed to a simple $k_{2D}(T)$ dependence with k decreasing with a temperature rise.

2. Inhomogeneous heating

Because of the device geometry, the Joule heating in the microheater is indeed spatially inhomogeneous. This is visible both in our COMSOL simulations and in experiments without the 2D heterostructure (thermal radiation maps, Fig. S6–18), as well as in Fig. 3 (with the 2D layer). We explicitly include and explain this inhomogeneity in all simulations, this is precisely why we rely on COMSOL for accurate modeling. Despite this, the inhomogeneous heating does not reproduce the nonlinear profiles, neither along the heater nor in the transverse direction (Figure 5). Raman thermometry shows no evidence of defects or local perturbations, and we obtain excellent agreement between simulated and measured temperature profiles along the microheater for HS1 and HS2. Therefore, inhomogeneous heating alone cannot account for the nonlinear behavior.

Figure S6-18: Parasitic thermal path with thermal radiation and thermal conduction through the residual air. a) Raman spectra of Silicon on the two cantilevers without suspended 2D material with and without Joule heating of the microheater with the corresponding temperatures extracted from the peak position. The different arrows indicate the measurement positions. b) Temperature map along the two cantilevers with Joule heating.

3. Contact variations

We also rule out contact-related artifacts for several reasons:

Persistence under strong gradients: The HS2 data in Fig. 4a were taken under very large temperature gradients. If inhomogeneous heating near the contacts were the origin of the nonlinearity, such effects should become more pronounced at higher heating power. This is not observed.

Reversibility: The nonlinear behavior is fully reversible when reversing the heat flow, which is inconsistent with a fixed faulty contact or asymmetric interface resistance.

Agreement with simulations for HS1: In Fig. 3 (HS1), the measured profile matches simulations that assume homogeneous interfacial contact. This indicates that contact quality is uniform in that device, and there is no reason to expect poorer contact for HS2 or HS3.

Multiple spatial measurements: Thermal contact resistance is negligible in our configuration, and temperatures are extracted from multiple points along the contact region, including transverse measurements. These consistently show no signature of inhomogeneous contact resistance.

Transverse profiles contradict a contact-based explanation: In Fig. 5c (HS3), the transverse temperature profile is almost flat near the hot electrode, even though the contact area there is small, while it varies by more than 100 K near the cold electrode where the contact area is much larger. This trend is the opposite of what would be expected if contact inhomogeneity played a dominant role.

5- Claims of “highest conductivity” and “robust evidence” are not justified given the uncertainties and lack of validation.

Answer: We modify the text in accordance to the comment but we justify the term highest conductivity afterwards

For the term highest or higher, when related to the measured conductivity of Hbn, we modify “The thermal conductivities exceed 1650 W·m⁻¹·K⁻¹ at room temperature, significantly higher than in previous reports” by “The thermal conductivities exceed 1650 W·m⁻¹·K⁻¹ at room temperature, significantly high compare to previous reports”. We modified “From these measurements, we

extrapolate higher values for the thermal conductivity of hBN than those previously reported” by “**From these measurements, we extrapolate high values for the thermal conductivity of hBN**”. We replaced “As shown in Figure 3d, our value for k_{2D} of about $1650 W \cdot m^{-1} \cdot K^{-1}$ represents the highest reported to date for hBN” by “**As shown in Figure 3d, our value for k_{2D} of about $1650 W \cdot m^{-1} \cdot K^{-1}$ represents a high value of thermal conductivity**” and we modify “the response of the hBN material due to its much highest conductivity, relative thickness, Young Modulus and bending stiffness compare to WSe₂” by “**...the response of the hBN material due to its much highest conductivity, relative thickness, Young Modulus and bending stiffness compare to WSe₂**”

For the term robust, we have replace “Using a rigorous temperature calibration and a robust measurement methodology” by “**Using a rigorous temperature calibration, we demonstrate ...**” and also we have replaced “These results are robust across different measurement conditions” by “**These results are obtained across different measurement conditions**”.

For the highest conductivity, our value is $k_{2D} = 1650 \pm \frac{550}{350} W \cdot m^{-1} K^{-1}$. Let consider the uncertainty in the rest of the litterature, at less for the high value of thermal conductivity (ref 23,24,61 and 42 in the SI):

For (Mercado, E., Yuan, C., Zhou, Y., Li, J., Edgar, J. H., & Kuball, M. (2020). Isotopically enhanced thermal conductivity in few-layer hexagonal boron nitride: Implications for thermal management. ACS Applied Nano Materials, 3(12), 12148-12156.) They measure $k_{2D} = 630 \pm \frac{90}{65} W \cdot m^{-1} K^{-1}$.

In (Yuan, C., Li, J., Lindsay, L., Cherns, D., Pomeroy, J. W., Liu, S., ... & Kuball, M. (2019). Modulating the thermal conductivity in hexagonal boron nitride via controlled boron isotope concentration. Communications physics, 2(1), 43.) They measure $k_{2D} = 585 \pm 80 W \cdot m^{-1} K^{-1}$ at ambient temperature.

In (Cai, Q., Scullion, D., Gan, W., Falin, A., Zhang, S., Watanabe, K., ... & Li, L. H. (2019). High thermal conductivity of high-quality monolayer boron nitride and its thermal expansion. Science advances, 5(6), eaav0129.) They measure $k_{2D} = 751 \pm 340 W \cdot m^{-1} K^{-1}$ at ambient temperature.

In (Cai, Q., Scullion, D., Gan, W., Falin, A., Cizek, P., Liu, S., ... & Li, L. H. (2020). Outstanding thermal conductivity of single atomic layer isotope-modified boron nitride. Physical Review Letters, 125(8), 085902.) They measure $k_{2D} = 1009 \pm 313 W \cdot m^{-1} K^{-1}$ at ambient temperature.

Only the last publication measurement is overlapping lightly with our measurement at ambient temperature. At lower temperature the second publication obtained $k_{2D} = 1300 \pm 200 W \cdot m^{-1} K^{-1}$ at 140 Kelvin and can be compared to the value obtained here.

We modified the text and add the four publications in the main text.

As shown in Figure 3d, our value for k_{2D} of about $1650 \pm 550/350 W \cdot m^{-1} \cdot K^{-1}$ represents a high value of thermal conductivity compare to the literature with $k_{2D} = [630 \pm 90/65, 585 \pm 80, 751 \pm 340, 1009 \pm 313] W \cdot m^{-1} K^{-1}$ 6,5,7,8.

Perhaps presenting anti-Stokes/Stokes temperature maps alongside shift-derived results would help substantiate the conclusions.

Answer: We tried to use Antistokes and stokes measurements to confirm our result. In fact, this methods require a calibration on its own as shown in suplementary information section S3, Figure S3-9. This method require also more time for integration due to smaller signal of the Antistokes signal compare to the Stokes one, especially at low temperature. We did not measure temperature map with Stokes/AntiStokes. We will explore this in the futur with a much more stable system (the Linkam cell that has been used is limited for such applications due to the limited time of meaurement without deviation/few hours, we propose to operate in a new helium free cryostat)

The manuscript would also be more convincing if the authors could demonstrate that the Raman probe laser does not introduce measurable heating.

Answer: We fully agree with the reviewer. Experimentally, we take care of this laser heating process. One motivation is to have the best signal on noise ratio without additionnal heating. It is well reference in the litterature and we already measured some thermal conductivities with laser heating (for an

example among other Chaste, J., Missaoui, A., Huang, S., Henck, H., Ben Aziza, Z., Ferlazzo, L., ... & Ouerghi, A. (2018). Intrinsic properties of suspended MoS₂ on SiO₂/Si pillar arrays for nanomechanics and optics. ACS nano, 12(4), 3235-3242.) Basically, with a confocal laser power at 532 nm around 50 μ W for a spot size below 1 μ m, it is normal to observed laser heating of suspended monolayer 2D material. For the thick sample measured in the paper, we did the calibration before measurement and a result can be see below Figure S4-11. We do not observe a laser heating of hBN up to 1.6 mW for silicon and WSe₂ and 32 mW for hBN

We modified the text in order to clarify this point. We add here a complet set of measurement of Raman spectroscopy in function of the laser heating for each material in a suspended heterostructure

Figure S4-11: Raman signature of a suspended heterostructure with WSe₂ on top of hBN as a function of the applied optical power. (right) Raman signature of Si microheater as a function of the applied optical power. (middle) Raman signature of WSe₂ on top of the hBN as a function of the applied optical power P. (left) Raman signature of the hBN as a function of the applied optical power.

We do not observe a laser heating of hBN up to 1.6 mW for silicon and WSe₂ and 32 mW for hBN. The power used for most of the measurement is around 320 μ W, so far from any Raman heating signature. It is in accordance with previous result obtained with the same set-up.

Cross-validation with an independent thermometry technique would further strengthen the work.

Answer: We fully agree that cross-validation with an independent thermometry technique would further reinforce our conclusions. However, this question is intrinsically related to a central point of our study: the thermal conductivity of 2D materials is strongly dependent on the measurement technique and on the measurement geometry. In particular, suspended and supported samples yield markedly different values. Most of techniques such as thermoreflectance or SThM are typically restricted to supported samples. Moreover, the extracted thermal conductivity depends not only on the sample dimensions (length and thickness), but also on the characteristic length scale of the temperature gradient (for instance the laser spot size in optical methods). For these reasons, publications combining multiple techniques on the same 2D system remain relatively rare.

We are currently performing preliminary SThM measurements in collaboration with Séverine Gomez at CETHIL (Lyon University, France) on samples suspended over similar microheaters. These early results tend to confirm the values reported in the present work, although more experiments are needed. Because SThM on suspended 2D materials is itself an emerging topic, we anticipate that these results may lead to a separate publication.

It is also important to emphasize that the temperature gradients in our device are unusually large (up to 250 K between suspended microheaters). Many conventional techniques cannot operate reliably under such conditions. For example, we previously used optomechanics (Chiout et al., npj 2D Mater. Appl. 7, 20 (2023)) and Raman heating+thermometry (Chaste et al., ACS Nano 12, 3235–3242 (2018)) to

measure thermal conductivity in suspended 2D samples, but these approaches are not applicable here. Similarly, our collaborative low-temperature thermoelectric measurements (<https://arxiv.org/abs/2507.03436>) probe a completely different regime ($\Delta T < 5$ K).

Nevertheless, we performed an independent cross-check using modulated thermoreflectance in Section 13 of the Supplementary Information. This technique was used to measure the in-plane thermal conductivity of isotopically enriched hBN flakes. We observed the expected increase in thermal conductivity compared with natural hBN, with extracted values G_{th} and $k_{hBN//}$ were approximately 47 $MW \cdot m^{-2} \cdot K^{-1}$ and 470 $W \cdot m^{-1} \cdot K^{-1}$.

Comparable thermoreflectance measurements on non-isotopic hBN flakes of slightly lower thickness (30–40 nm) yield smaller values (~ 250 $W \cdot m^{-1} \cdot K^{-1}$), consistent with expectations.⁹

Finally, we note that the difference between Raman- and thermoreflectance-based values is broadly consistent with the well-known factor-of-two-to-three discrepancy between suspended and supported geometries. As shown in Figures 2–3 and in Supplementary Section 12, such variations in boundary conditions naturally produce variations of this magnitude, in line with our measurements.

Also, we justify our work and our technic of thermal conductivity measurement with the measurement of a graphene sample which is a reference. Our value of conductivity is 2800 $W \cdot m^{-1} \cdot K^{-1}$ which is in accordance to the literature. It certified the validity of our technic for hBN.

In addition, a full parameter sensitivity analysis of the COMSOL fitting would be valuable.

Answer: This is equivalent to the question 3, please comeback to the answer.

We add in inset of Figure 2a and 2b these points for the sensitivity measurements in Figure 2a and 2b. and add in the caption “In inset the sensitivity S as $S = \partial T / \partial k$ ”

We demonstrate in the supplementary information, that the measured temperature profile is not affected by residual strain (S5), laser probe heating (S4) and we proceed to analyses and discussion, with uncertainty analyses, (with a graphic approach of the sensitivity definition) along the supplementary information for microheater and 2D thermal conductivities (S6, S8), temperature dependence of the thermal conductivity (S9) the hBN out-of-plane thermal conductivity (S10), interfacial thermal resistance (S11).

Finally, the authors are encouraged to quantitatively assess whether diffusive models, incorporating realistic parameter variations, could reproduce the observed profiles.

Answer: We take also this question as equivalent to the question 3, please refer to the answer above.

REVIEWER COMMENTS

Reviewer #1 (Remarks to the Author):

The authors' detailed response letter addressing the review comments is greatly appreciated. The revised manuscript indicates a significant improvement on clarification of novelty, and a better narration on the experimental results and discussions. Overall, I recommend the manuscript for publication, subject to the following minor revisions.

1. In the discussion of Figure 3, the authors clarified the 6-point fitting strategy by mentioning "increasing the number of sampling points does not improve the fitting quality". It would be good if evidence (maybe a figure comparing different fitting strategies) could be provided in the supplementary material to support this.

Answer: We thank the reviewer for this constructive suggestion. To address this point and provide quantitative support for our statement, we have added an additional analysis to the Supplementary Information comparing fitting strategies using different numbers of sampling points.

In general, a two-point configuration is sufficient only when thermal contact resistances are negligible. When the thermal contact resistance becomes comparable to the intrinsic thermal resistance of the sample, at least a four-point measurement is required to properly constrain the temperature profile. In the case of sample HS1, a four-point thermal measurement revealed a small but systematic deviation, of only a few kelvin, between the measured temperature and the simulated temperature map at the coldest region of the silicon microheater (points T1 and T6 in Fig. 3a). This motivated the use of a six-point configuration, which provides a temperature map that is much closer to the experimental reality.

As shown in Figs. 3a–c, the temperature distribution obtained from the six-point analysis already reproduces the experimental data extremely well. Importantly, adding additional sampling points will not lead to any visible improvement in the quality of the fit.

Figure S16: Standard deviation between experimental temperatures and simulated temperature for sample HS1 in function of the hBN thermal conductivity and the Silicon thermal conductivity. We plot the result for $N=2,4,6$ and 11 points of temperature. The white part correspond to uncertainty above 2 degrees.

To quantitatively justify this conclusion, we now include an additional figure in the Supplementary Information showing the standard deviation (in kelvin) between the experimental data and the fitted temperature map as a function of the number of sampling points ($N = 2, 4, 6,$ and 11).

In this analysis, the thermal contact conductance was fixed to the value extracted independently (see Section S11 of the Supplementary Information). This explains why the two-point configuration can still yield a reasonable result in this specific case, as the contact conductance is not treated as a free parameter. Indeed, the thermal contact conductance cannot be reliably determined from a two-point measurement alone.

We observe that the minimum standard deviation is reached for both the four-point and six-point configurations, with a slightly shift of the minimum standard deviation, i.e the best solution. Between the four-point and six-point analyses, the optimized thermal conductivities of hBN and silicon differ by approximately $100 \text{ W}\cdot\text{m}^{-1}\cdot\text{K}^{-1}$. This difference originates from the improved constraint provided by the additional points located farther from the heater in the six-point configuration and a better fit of our whole data. However, this correction remains relatively small.

A similarly but smaller shift in the optimized parameters is observed when increasing the number of points from six to eleven, again on the order of less than $100 \text{ W}\cdot\text{m}^{-1}\cdot\text{K}^{-1}$ for the hBN thermal conductivity. This seems negligible and this confirms that the six-point configuration already captures the essential physics and provides a robust and accurate fit. Even the 4 point measurement seems accurate from this perspective.

Finally, the apparent increase in the minimum standard deviation with the number of sampling points is a generic fitting effect: while a two-point dataset can always be fitted perfectly, this is no longer true

standard deviation is null for $N=2$

standard deviation has a minima >0 for $N>2$

when fitting three or more points, as can be explained schematically on the Figure below. This behavior is analogous to fitting a straight line through two points versus three or more points.

Based on this systematic analysis, we conclude that using more than six sampling points does not significantly improve the fitting quality, while increasing experimental and computational complexity. The six-point configuration therefore represents an optimal compromise between accuracy and efficiency.

We added this discussion to the supplementary information section 16

2. The sample nomenclature (HS1, HS2, Gr1) is updated in the main text, but not in the supplementary information. It would be good to make them unified.

Answer: We corrected the supplementary information document and sample names are now uniform through all the documents.

Reviewer #2 (Remarks to the Author):

The revised manuscript addresses most of my concerns in a satisfactory manner. The authors have clarified the experimental methodology, improved the presentation of the multi-point thermal measurements, and significantly moderated their claims regarding non-diffusive heat transport. The exceptionally high thermal conductivity values are now better contextualized through control experiments and a clearer discussion of sample geometry, isotopic purity, and suspension effects. While some interpretations, particularly regarding the validity of Raman thermometry under strongly non-diffusive conditions and the precise physical origin of the observed non-linear temperature profiles, remain suggestive rather than conclusive, the authors now appropriately acknowledge these limitations and avoid overstatement. Overall, the work presents a technically innovative experimental platform and compelling phenomenological observations that will be of broad interest to the thermal transport and 2D materials communities. I therefore recommend publication after minor revision, provided that the cautious framing of the conclusions is maintained throughout the manuscript.

Answer. We thank the reviewer for this thoughtful and positive evaluation of our revised manuscript. We appreciate the constructive feedback, which has helped us to clarify the methodology. We believe these changes have significantly improved the overall quality and clarity of the work.